# Dingoes, companions in life and death: The significance of archaeological canid burial practices in Australia

Loukas George Koungoulos[1]*, Jane Balme[2], Sue O'Connor[1]

1 School of Culture, History and Language, College of Asia and the Pacific, The Australian National University, Canberra, Australia, 2 School of Social Sciences, University of Western Australia, Crawley, Australia

* loukas.koungoulos@anu.edu.au

## Abstract

The dingo, also known as the Australian native dog, was introduced in the late Holocene. Dingoes were primarily wild animals but a number resided in Aboriginal people's camps. Traditionally, these individuals were taken from wild litters before weaning and raised by Aboriginal people. It is generally believed that these dingoes were not directly provided for, and upon sexual maturity, returned to reproduce in the wild. However, some died while in the company of people and, were buried in occupation sites. This Australian practice parallels the burial of domestic dogs in many regions of the Asia-Pacific and beyond but has attracted very little research. We explore the historical and archaeological evidence for dingo burial, examining its different forms, chronological and geographic distribution, and cultural significance. Dingoes were usually buried in the same manner as Aboriginal community members and often in areas used for human burial, sometimes alongside people. This practice probably occurred from the time of their introduction until soon after European colonisation. We present a case study of dingo burials from Curracurrang Rockshelter (NSW) which provides insights into the lives of ancient tame dingoes, and suggests that domestication and genetic continuity between successive camp-dwelling generations may have occurred prior to European contact.

## Introduction

The dingo is an Australian canid of debated taxonomic identity, presently commonly referred to as a long-term feral variant of primitive domestic dog *Canis familiaris* [1, 2], and otherwise as a unique species *Canis dingo* [3, 4]. Dingoes first appear in the Australian fossil record in the late Holocene, with the two oldest available direct dates being from the Nullarbor Plain at the southern edge of Australia. Direct AMS dates from dingo fossils of 3363–3211 cal. BP (3069 ± 27 SANU 54821) for the deepest bones found in Madura Cave [5], and 3259–3022 cal. BP (3031 ± 34 OxA-27532) for a specimen from surface deposits in Koonalda Cave [6] provide a minimum entry time for dingoes to the Australian continent of approximately 3300 years

**Funding:** J.B. and S.O. received an Australian Government Australian Research Council (https://www.arc.gov.au/) Discovery Project award

(DP210101960) which funded this research. The funders had no role in study design, data collection and analysis, decision to publish, or preparation of the manuscript.

**Competing interests:** The authors have declared that no competing interests exist.

before present. A late Holocene arrival timeframe best fits the evidence from New Guinea and other islands to Australia's north where dog bones are not found in contexts older than 3,300 years BP [7]. However, more direct dates are needed, especially from northern Australia where few dates have been obtained, to refine the timing of arrival and the rate of dispersal across the continent.

Previous suggestions about the rapidity of dispersal around the continent by comparing dates on dingo remains from across the continent with the historically documented dispersal rate of other introduced predators such as foxes and cats [8–11] have argued that dingoes' dispersal around the continent was rapid–perhaps a few hundred years [5, 12, 13]. These comparisons may not be precise analogues for dingo because, unlike dingoes, there were multiple releases of foxes and cats in different locations [14]. However, to reach Australia dingoes had to have accompanied people on watercraft and it is thus very likely that they arrived in a tamed state if not domesticated state. This is because there is no evidence that populations of wolves or wild dogs akin to dingo existed during prehistory in locations proximate to Australia which could have served as their source [15]. A potential exception are the wild dogs of New Guinea, but these are extremely closely related to dingoes, and were almost certainly introduced to New Guinea in a tamed or domesticated state at approximately the same time [16]. Dingoes arriving in a tamed state were probably taken up by Aboriginal people very quickly and this association could well have facilitated their movement across the continent over a relatively short period of time, as occurred with domestic dogs in contact-era Tasmania [17].

By the time Europeans settled in Australia the close bond between dingoes and Indigenous people was entrenched. This is well known by Indigenous people and has been documented by numerous observers. In daily life dingoes were used for a variety of purposes including as personal protection, warmth, companionship, as guards, and as a "living technology" for hunting [18–22]. These uses, including the degree to which dingoes were used as hunting aids, varied geographically [7]. A variety of written, oral and visual sources suggest that dingoes were incorporated into Indigenous people's kin systems and cosmologies, holding important positions in stories and being the subject of songlines and ceremonies across vast areas of Australia [23–27]. Dingoes and dingo tracks also appear as rock art motifs, especially in northern Australia [28, 29].

It is widely recorded that camp dingoes were gathered as pups from dens and brought back to camp where they were reared by people [21, 22]. A common belief is that once the camp dog reached sexual maturity it returned to the wild. For example, Lumholtz, speaking about northern Queensland, believed that dingoes were never domesticated because they often run away from camps, especially in the "pairing season" and at such times never returned [30]. Thomson states that "the dingo seldom bred in captivity but after a period of perhaps two or three years of life in a native camp he went into the bush to breed" [24], although sexual maturity is usually reached between one and two years of age. Later, Macintosh suggested that on reaching maturity a tame dingo would "simply take off and return only to steal food" [31]. Naturally, some tamed dingoes would die before this occurred. The practice of Aboriginal people then providing burial rites for these dingoes (and later dingo hybrids and European dogs) was recorded by Europeans during the early stages of settlement in different parts of Australia. The nature of these rites varied across the continent and in many places matched those provided to humans.

The general scholarly consensus is thus that because colonial-era dingoes left Aboriginal camps to reproduce in the wild, they should not be considered "domesticated" at least during their history in Australia [32]. Ballard and Wilson [33] alternatively consider draw a line between traditional domestication as the end product of intentional artificial selection and "taming" achieved through unconscious, non-directed anthropogenic selection, framing dingoes in the company of Aboriginal people in past and present as "tamed" and modern wild dingoes as "untamed".

Yet these contentions are primarily if not entirely based on the study of the biology, genetics and behaviour of modern dingoes, from 1788AD onwards, and there is a distinct possibility that different domestic arrangements existed between humans and dingoes in the time before the major, permanent disruptions caused by European invasion. Previous studies have drawn attention to dingo remains found within Aboriginal habitation sites, most of which appear to be the product of deliberate burial by people in pre-contact times, which are suggestive of longer-term associations [22]. The most important of these are Gollan's appraisals of ten archaeological dingoes from southeastern Australia, which found in many individuals aspects of dental attrition suggesting reliance on provisioning over the course of their lives [12, 34]. Fish remains were present in the stomach cavities of two of these individuals (Kioloa 1 and Mallacoota 1) supporting this claim. Of the 10, seven were estimated to be more than six months old with both sexes represented [22]. Furthermore, some exhibited unusual morphological features that were suggestive of altered selection and potentially restricted breeding pools occurring in relation to anthropogenic influences [12, 34].

Dingo burials are thus an invaluable archaeological resource with great potential to inform our understanding of the pre-contact relationships between people and their dingoes, which is heavily reliant on projections made from historical ethnographic data [19–21, 32], the obvious issue being that such information may not be sufficiently detailed or relevant to explain aspects of pre-contact human-dingo interactions. Despite this, dingo burials remain and largely understudied outside of Gollan's studies, which had a biological-evolutionary focus and were primarily concerned with using burial remains as a source of pre-contact morphological data [12, 34]. Compared to broad-scale knowledge of Australian burial practices for people, very little is known about the geographic distribution, depth of antiquity, or modes of burial utilised for dingoes; and very few have received osteological analyses that may information about demography, life-history and interactions with people.

In this paper we comprehensively survey the historic and archaeological evidence for dingo burials and what these sources reveal about the relationship between First Nations people and dingoes, with attention to the manner and modes of burial, and their collective chronological and spatial distributions. We also present an archaeological case study of dingo burials from the site of Curracurrang Rockshelter, an occupational site in southern coastal New South Wales, which focuses on osteological indicators of the life-history of these individuals and their likely interactions with Aboriginal people. Our study considerably expands understanding of a crucially important but historically obscured aspect of First Nations-dingo interactions in precolonial Australia, which has particular importance to the history of dogs and their domestication in the greater Asia-Pacific region.

## Methods

Our study pursued three avenues of investigation. First, we examined historical and ethnographic texts dating from the beginning of colonial history in Australia (1788 AD) to ascertain the nature of dingo burial as it occurred within the last 200 years. We relate the salient aspects of accounts recorded by Europeans concerning dingo burial, and later the burial of dingo-hybrids or domestic dogs, with particular attention to variation in the modes of burial employed between different regions and how the burial mode employed for dingoes/canids compares to the mode(s) used for human members of the same societies. This forms an important and crucial background to the next two avenues, which focus on archaeological evidence.

We then surveyed the occurrence of dingo burials in Australia in the archaeological record, as reported in archaeological literature and as news stories, relating information concerning geographic occurrence, indications of antiquity, the mode of burial and whether human

remains were also present. It is not always easy to recognise deliberate burial of dingo individuals. Most dingo are interments, which begs the question of how they can be recognised as a deliberate burial rather than the remnants of people's meals, or a naturally-preserved death of an unassociated (wild) individual within what happens to be a cultural site. The major criterion indicative of deliberate burial is the presence of relatively complete, articulated dingo skeletons found within archaeological sites. Dingoes were eaten by many Aboriginal groups [15], but if the bones of such meals were discarded at site as food refuse, the skeleton should be incomplete, disarticulated, scattered, and possibly bear evidence of butchery or cooking as is apparent for other mammalian dietary species of comparable size.

In some sites there is evidence of cuts through the stratigraphy delineating the grave edges (e.g. Mallacoota) [34]. However, in shell middens and rock shelters, such as those on the south east coast of New South Wales, where the stratigraphy is unclear or where the burial was recovered without recording stratigraphic detail, the facts that complete skeletons are present and that they are within an occupation site are appropriate criteria for recognising deliberate burial. If the dingo had died in a rock shelter without burial, it would be exposed to other scavengers, including other dingoes and reptiles. These attacks would result in the disarticulation of the skeleton, removal and scattering of bones. In open site contexts, the carcass of a dingo dying without burial would disintegrate as a result of exposure unless there were environmental mechanisms for covering the bones with sediment, such as flooding or aeolian activity. Thus, in the absence of alternative explanatory mechanisms, the relatively complete and articulated dingo skeletons found within or in immediate proximity to archaeological sites are deemed to most likely be the result of deliberate burial by people.

Finally, we undertook a case study of buried dingo remains from the site of Curracurrang, a rockshelter habitation site southern coastal New South Wales (see Fig 1). We determined the stratigraphic context of dingoes in the Curracurrang deposit using original excavation recording notes as well as published materials, and also obtained direct AMS dates on the bones of several individuals. Together these provide a chronology for the earliest directly dated dingo burials in southeastern Australia, if not the continent as a whole. We also employed a generalised osteological assessment for the dingo remains from this site, examining each individual variably, as permitted by completeness and preservation condition for information concerning morphology/phenotype, individual age, diet, and pathologies that inform the life history of individual dingoes and what these may suggest about their interactions with Aboriginal people in pre-contact Australia.

## Results

### Historical perspectives on dingo and dog burial

Historical sources relating information about First Nations canid burial practices are concentrated in northwestern and southeastern Australia, in an interesting parallel to the distribution of their occurrence archaeologically (see below). W.G. Stretton, who lived in the Northern Territory while holding various positions including in the police, postal and customs departments between 1865 and 1920 [35] reported that in the western Gulf of Carpentaria around Roper River, people "always have a lot of dingoes with them. . .and are quite as fond of their dogs as they are of their children" [36] Because Stretton initially refers to dingoes here, it is clear that his subsequent use of the word dog refers to dingoes. "They also put the dead dogs into the branches of a tree on a stage, but after that pay no attention" [36].

This disposal method mimics part of the rites that he also reported for people in the area who, after death, were wrapped in paper bark and placed on a platform among the branches of a tree [36]. The body remained on the platform until the flesh was no longer on the bones

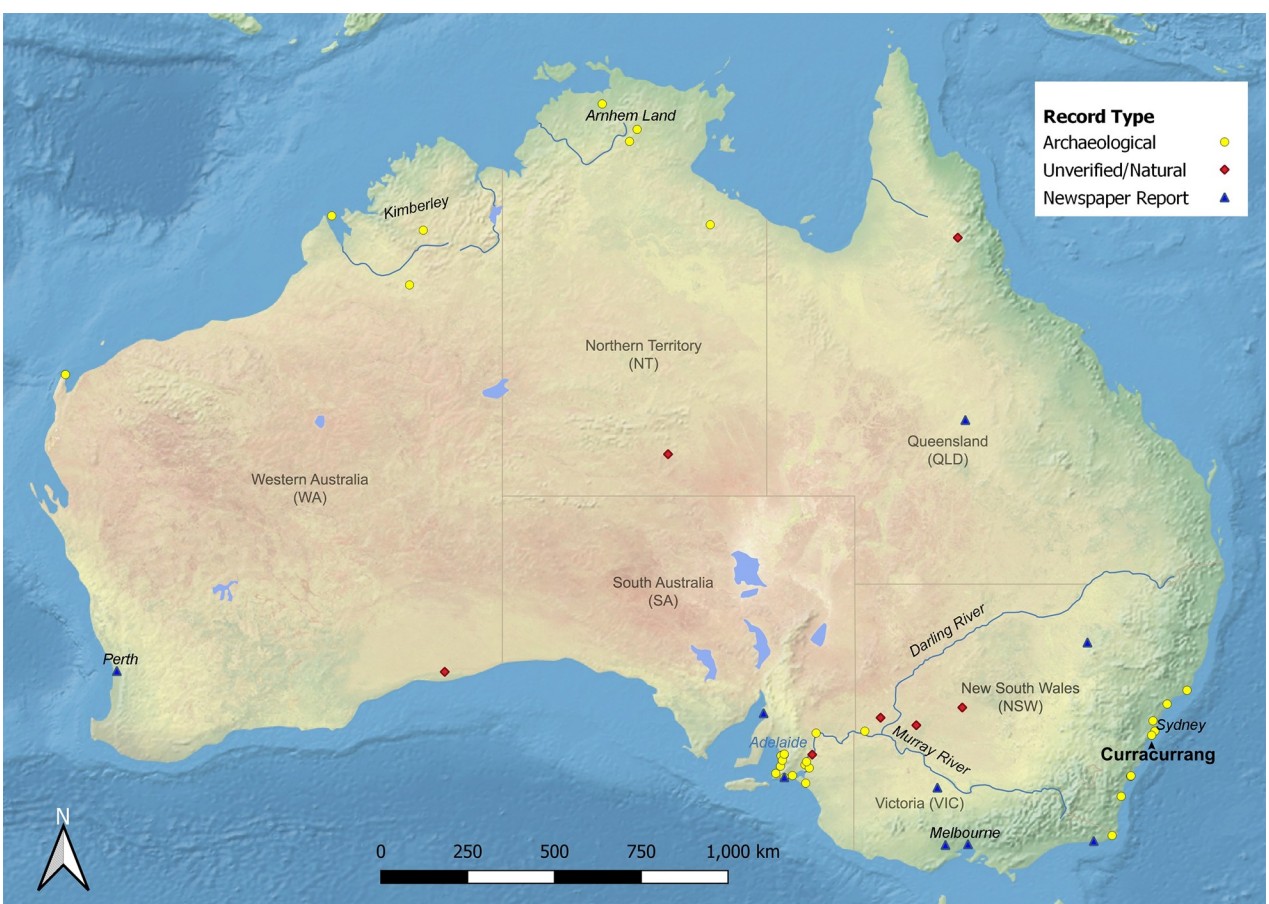

**Fig 1. Distribution of archaeological dingo and dog burials, including unverified pseudo-burials and cases reported in news media.**

which were then collected and placed in a hollow log. The log was then placed in branches of a tree—usually one that was difficult to access, such as in a rocky ravine [36]. In these descriptions Stretton uses a male pronoun "his body" and "his relatives" so it is not clear whether this process was followed only for men or for all the community [36].

In the Keep River region, adjacent to the Northern Territory/Western Australia border, Mulvaney reports that rockshelters were used as wet season residences, for housing burials and as places to cache select possessions [37]. He was told by Mirriuwung and Gadjerong Traditional Owners that the remains of favoured dogs were buried in the same way as humans–in bundle burials and then placed in rock shelter clefts [37]. It seems here that the Traditional Owners are referring to a past, rather than current practice.

In Worrorra country in the Western Kimberley, Rev. J. Love worked on Presbyterian missions between 1914 and 1915, and again between 1927 and 1941. He had a genuine interest in the people and, as well as learning the language to some degree, he recorded various aspects of the lives of the Worrorra including their burial practices [38–40]. Men were given compound burials which included placing the body on a wooden platform. Once the flesh had dried, the leg-bones were wrapped in one parcel and the remaining bones in another with the larger parcel being placed in a cave. The smaller parcel was carried around for as long as a year before also being placed in a cave. Women's bodies, on the other hand, were placed without ceremony in a rock crevice and then covered with stone, but before disposal, the flesh of the upper part of the

body was eaten [39]. Very young children were buried with a layer of stones placed on top [40]. However, Love's accounts are somewhat inconsistent as in 1936 he says that the corpses of children were wrapped in paper bark and carried before being deposited in a cave but elsewhere says that the bodies of young people were placed on platforms in tree branches [38, 39].

Love also described the close relationship between people, especially women, and dingoes. "Every woman had several dingoes, either born to her tame dingoes, or captured in the bush as wild puppies and tamed" [39]. He describes an incident where he shot a dingo owned by a woman who "with loud shrieks and wails, carried it off for burial up in a tree". An image of one such tree burial has the accompanying caption that: "Tree burials were used for both pet dogs and young people" [39]. Love wrote that "dogs are given names and are accorded a more pretentious funeral than falls to the lot of an old women. The dingo is wailed over in the same way as a man, placed in the branches of a tree, covered with a sheet of paperbark and there left to fall to pieces" [38].

In central Australia today, Warlpiri communities live with dogs, call them by kinship terms, and speak of them as family (*marlpa*) and companions (*warlalja*) [41]. Although Warlpiri acknowledge dogs as not human, they place dogs on the "human", morally good end of their personhood spectrum ranging from human—stranger—inhuman—monstrous [41]. While they do not cry for every dead dog–this depends on the dog, its history, and its human network–any camp dog that dies, gets buried. Presently both people and dogs are buried in the ground, but in the past Warlpiri people practiced primary burials on tree platforms and dogs were placed on top of bushes [41]. Interment burials for the Warlpiri originated during the mid-20[th] century mission period, but "bush burials" for dogs were still used in the early 21[st] century, with the transition to interment occurring at some point during her ~20 years of fieldwork [41].

Meehan and colleagues' work, based largely on Meehan and Jones' observations in the early 1970s, report that the Anbarra people of western Arnhem Land distinguished between "rubbish" dogs and "good" dogs, the latter being those that were good hunters or general companion dogs [42]. They did not see dogs being cremated or placed in caves, instead the Anbarra had begun to bury their dogs in graves–at least one of these dogs was wrapped in paper and cardboard before burial [42]. At the time of their fieldwork report European breed dogs were particularly favoured by Anbarra people, as the practice of collecting dingo pups from the wild had ceased, but at least one camp dog had a dingo father [42]. Thus in the recent Anbarra context it is likely that buried dogs were European breed dogs, mixed-breed camp dogs and (less frequently) dingo-dog hybrids.

Cahir and Clark [43] have summarised primary historic sources about the relationship between British colonisers and Aboriginal people in Victoria. They cite Samuel Rawson, a squatter southeast of Melbourne in the mid-nineteenth century, who stated he shot some dogs in retribution for the dogs killing his poultry. Rawson noted in his diary the calamitous effect this had on the dogs' Aboriginal owners: "they buried the dead bodies of their four legged companions with great ceremony, wrapping them in blankets and sheets of bark & lighting fires by their graves after which they decamped & moved up the river" [43]. This is affirmed by William Thomas, an Assistant Protector of Aborigines of Port Phillip at the time, who recorded that Victorian Aboriginal people performed mortuary ceremonies for their dogs [43]. Because the area of these records is close to a British colony, it is possible that the dogs in question were not pure dingoes.

However, the burial procedures for these dogs match those described for the region in Meehan's review of the historic and archaeological evidence for mortuary practices in Australia which indicate that burial was the disposal method for people [44]. She cites Howitt [45], Morgan [46] and William Thomas [47]. However, Howitt is citing Dawson, who describes several Victorian disposal methods, including cremation but says that people "of common rank" are

wrapped in a possum skin rug and then placed between two bark sheets before burial [48]. Morgan's observations are near the Barwon River and, while he does record the burial of two women in circular graves over which upright sticks are inserted, he also describes the disposal of a man who was rolled up in his skin "rug" and then put on a platform placed in a tree and covered with bark to protect from birds of prey. This observation may indicate a practice peculiar to the Barwon River area. Thomas earlier described the burial of an important man but does not provide details of the process except that two "favourite" dogs were killed, bound and burned as part of the funeral ceremony [47].

In describing funeral practices of the Melbourne area in 1840, Howitt says that people were interred in the ground, and that he came across one such grave in which was a circular mound of earth about a foot high, around which were driven 20 stakes. However, subsequently he says that women and children, if buried at all, are dismissed unceremoniously [49]. This seems to contradict Morgan [46], whose descriptions of women's burials fit that of Howitt's earlier description [45].

It is evident that variations in funerary and burial traditions existed within many traditional Aboriginal societies, some of which may have corresponded to demographic and social characteristics of the individual in question. Nevertheless, according to Meehan's survey it is clear that a single-stage interment in the ground was the predominant basic form of burial practiced throughout much of southeastern Australia [44].

These examples establish that at least in historic times, companion dingoes/dogs were mourned and given burials. While Meehan and co-authors describe both male and female "special" dogs as being afforded complex burials, the sex of the earlier reports that would have been about dingoes are not stated [42]. The other noteworthy aspect of these observations is that in all cases the burial rites given to "dogs" of whatever kind followed those rites given to people and that, as human rites changed after the arrival of Europeans, those of the companion canids did too.

## Archaeological evidence of dingo burials

Table 1 lists the dingo burials that have been reported in archaeological contexts, with their geographic distribution shown in Fig 1. We have indicated those records that do not clearly meet the criteria outlined in the methodology above. In all known cases the method of disposal of the dingoes is the same as those recorded in early literature for people in the same area, following Meehan [44]. Instances classified as unverified are suspected as being natural occurrences of "burial". These are concentrated in the lunette-rich lower Darling River environs, with singular occurrences in north Queensland and central Australia.

Before interpreting the results, it is important to first address a small number of complete dingo skeletons found within archaeological sites that also include human burials. In most of these contexts the dingoes seem to have been interred within dune sediments when the burrows (appropriated from wombats) they were inhabiting collapsed, sealing the remains within until revealed by wind erosion [50]. It is possible that tame dingoes were interred within empty burrows by people upon death, as some historical discoveries of dingo-human co-burials were located within rabbit warrens (Table 1), suggesting the occasional use of existing hollows for burial as a matter of convenience. However, as articulated skeletons of wombats and other burrowing species like bettongs (Potoroidae) are also found in the same localities within identical "collapsed burrow" contexts a non-cultural deposition is more likely. It is also possible that articulated dingoes are preserved in dunes if they were rapidly covered by wind-blown sediments soon after death. A similar mode of preservation is also possible at low-lying riverside rockshelter sites if alluvial sediment naturally covered the carcasses of wild dingoes which

**Table 1. Archaeological dingo and dog burials reported in published archaeological literature and news media.** Blank cells indicate lack of data.

| Site Name | Region | No. of burials | Context | Burial mode | Dating | Source | Comments |
|---|---|---|---|---|---|---|---|
| Broughton Island | Central coast, NSW | 1 | Exposed in coastal midden. | Buried in midden | 440±180 or 56–689 calBP [SUA-402C] human bone | Wright 1975:20 [55] | Two human burials recorded from same deposit. Lower end of calibrated range allows possibility of modern dog |
| Swansea Channel | Central coast, NSW | 1 | | Buried in midden | | Dyall 1972, 2012 [56, 57] | |
| The Meeting Place, Kurnell Foreshore Midden | Greater Sydney | 4 | Excavated from middens | Buried in midden | Dates for the midden range from 595–1951 calBP | Megaw [1968b] [58]; Tsoulos 2007:113 footnote and Irish 2010:18 for dates [59, 60] | Excavations by Megaw 1970–1971 and Irish 2008–2009. |
| Curracurrang | Greater Sydney | Several–see below | Excavated from rockshelter deposits | Buried in midden | Calibrated dates range from 726-2331BP See below | This paper, Megaw 1968a [61] | Variable completeness, age profiles |
| Apple Tree Bay, Kuringai | Greater Sydney | 2 | Excavated before 1911 from a rockshelter | Buried in midden | *radiocarbon date produced a 'modern' result interpreted as 18th or 19th century | Koungoulos 2022:317 [14] | These are likely to be hybrid or European dogs |
| Wagonga Inlet | South Coast, NSW | 1 | Excavated from base of coastal midden | Buried in midden | | Anderson 1890: 56–57 [62] | 'almost an entire skeleton of a small dog or dingo' |
| Kioloa (Nundera Point) | South Coast, NSW | 1 | Excavated from midden | Buried in midden | 937–1059 calBP (see below) | Gollan 1982: 162 Snelson et al. 1986:30 [34, 63] | Male |
| Murramurang | South Coast, NSW | 2 | 01 excavated from eroding midden Lampert 02 excavated from midden base. | Buried in midden | Basal date for 01 midden said to be <2000BP Charcoal from around burial 02 produced a modern date | Gollan 1982:142–145 [12, 34] | 01 Almust complete 'sub-adult'. Less than six months. ?female. 02 complete skeleton, mature adult male |
| Mallacoota (Captain Stevenson's Point) | Gippsland, VIC | 2 | 01 excavated from within midden 02 Incompletely excavated | Buried in midden | 01 below a radiocarbon date of 890±90 BP 02 c. 950–1300 BP | Gollan 1982:146–148 [12, 34] | 01 Complete skeleton in articulated position. Adult Male 02 Not all bone recovered. Juvenile estimated at 20 weeks |
| Mossgiel (Yarto Station) | Western NSW | 1 | No details given as to recovery method | Burial(?) or possible pseudo-burial | Uncertain, see comments | Anon. 1963 [64] | About 1 metre from one of two human burials. Associated with a date from one of the human bones. As no collagen was present, the minimum date of 6,000 years was obtained from bone apatite [65] As this was obtained in the 1960s and was not on dingo bone, suggestions as to the antiquity of the dingo remains must be regarded with caution. |
| Lake Milkengay | Western NSW | 1 | Excavated from deflating lunette | Pseudo-burial, naturally-preserved in collapsed den | Late Holocene dates obtained from charcoal laterally displaced by 10m, association dubious | Gollan 1982:132–133 [34] | Estimated age 1–1.5 years ?female |

(*Continued*)

**Table 1.** (*Continued*)

| Site Name | Region | No. of burials | Context | Burial mode | Dating | Source | Comments |
|---|---|---|---|---|---|---|---|
| Lake Mungo | Western NSW | 4 | 01 excavated from lunette 02 collected from deflated surface 04 excavated from deflated lunette 06 excavated from lunette | Pseudo-burial, naturally preserved in collapsed den | Found in probable Holocene wombat burrows dug into Pleistocene sediments, direct date for 01 forthcoming | Gollan 1982:139–141 [34] | 01 excavated from the Zanci stratigraphic unit which dates to 25-14kya [66] |
| Victor Harbour (the Bluff, Rosetta Head) | Lower Murray | 1 | | Buried in midden | | Pardoe 1996:3 [67] | Associated with 7 human burials and occupation debris. |
| Nunameena (Coorong Waters) | Lower Murray | 5 | | Buried in ground ["cemetery"] | | Pardoe 1996:3 [67] | Associated with 21 human burials and occupation debris. |
| Hindmarsh Island (Coorong) | Lower Murray | 1 | Excavated | Buried in sand | | Pardoe 1996:3 [67] | Associated with 1 human burial |
| Moana (Pedler's Creek) | Lower Murray | 3 | Excavated from coastal dune | Buried in midden | Direct date on dingo bone 1850±240 BP ETH2732 (2336–1299 cal BP) | Campbell 1988 [52] | Associated with occupation material |
| Greenfields Mound | Lower Murray | 4 | Excavated from mound deposits | Burials in occupational mound | Mound occupied from at least ~2000BP to c. 1850AD | Disspain et al. 2018; Draper et al. 2000 [68, 69] | Associated with 7 human burials and occupation material |
| Gillman Mound | Lower Murray | 1 | Excavated from mound deposits | Burial in occupational mound | Direct date 1774±20 NZA 35562 (250–396 AD) | Littleton et al. 2013:41 [53] | Associated with 22 human burials and occupation materials–although direct dates on human bone are younger |
| Dry Creek | Lower Murray | 1 | Excavated from mound deposits | Burial in occupational mound | | Pardoe 1996:3 [67] | |
| Glanville | Lower Murray | ≥1 | Excavated from mound deposits | Burial in occupational mound | | Pardoe 1996:3 [67] | |
| Ewell | Lower Murray | ≥1 | Excavated from mound deposits | Burial in occupational mound | | Pardoe 1996:3 [67] | |
| Kingborn Road, Parafield Gardens | Lower Murray | 2 | Excavated from mound deposits | Burial in occupational mound | | Pardoe 2015 [70] | |
| Swanport | Lower Murray | 1 | Excavated from midden or mound deposits | Burial in occupational mound | Human skeletons returned dates between 322–507 and 2955–2760 calBP | Pate et al. 2003; Stirling 1911:12 [71, 72] | Recovered amongst numerous human burials in a midden. Although some of the human bones were charred, or smoke affected, as is consistent with reports of smoking before burial in the area, no details of the dingo bones are provided- apart from the fact the skull was undamaged. |
| Brenda Park | Lower Murray | 5 | Excavated from rockshelter | Buried in ground | | Pardoe 1996:2 [67] | Associated with "family" of human skeletons |
| Fromms Landing (Tungawa) | Lower Murray | 1 | Excavated from rockshelter | Probable pseudo-burial | Between dates with calibrated ranges of 3569–3071 and 3340–2798 BP | Macintosh et al. 1964 [51] | Dingo skeleton found in culturally sterile layer in area of shelter unused by people, possibly was "buried" by flood sediments from river |

(*Continued*)

**Table 1.** (*Continued*)

| Site Name | Region | No. of burials | Context | Burial mode | Dating | Source | Comments |
|---|---|---|---|---|---|---|---|
| Normanville | Lower Murray | 1 | Excavated from coastal dune site | Buried in coastal midden [?] | | Koungoulos 2022:321 [14] | From "R" geological horizon |
| Tailem Bend | Lower Murray | 2 | Excavated from midden | Buried in midden/mound | | Koungoulos 2022:323 [14] | Occupational site with human burials |
| Mypolonga Wash | Lower Murray | 1 | Excavated or collected from sediments | | | Koungoulos 2022:322 [14] | |
| Victor Harbour (Whaler's Haven) | Lower Murray | 2–3 | Excavated from sandy deposit | Buried in sand | | Koungoulos 2022:321 [14] | Associated with occupational debris |
| Lake Victoria | Middle Murray | 1 | Recorded in situ in lunette | Buried in lunette | | Pardoe 1996:4 [67] | |
| Arnhem Land Plateau DB-1 | Top End | 1 | Exposed, examined in situ | Wrapped in paperbark and placed in rockshelter crevice | | Gunn et al. 2010 [73] | |
| Arnhem Land Plateau RWC-04 | Top End | 1 | Exposed, examined in situ | Wrapped in paperbark and placed in rockshelter crevice | | Gunn et al. 2012 [74] | |
| Dhua Lagoon | Top End | 1 | Exposed, examined in situ | Wrapped in paperbark and placed in rockshelter crevice | | Kamminga and Allen 1973:94 [75] | |
| The Dog Site | Top End | 1 | Exposed, examined in situ | Skull placed in rockshelter crevice | | Kamminga and Allen 1973:106 [75] | Rockshelter is an art site featuring dingo motif |
| McArthur River Station rockshelter | Top End | 1 | Exposed, examined in situ | Wrapped in paperbark and placed in rockshelter crevice | | Gunn et al. 2012:105 [74] | |
| Mordor Cave | Cape York | 1 | Excavated from cave deposits | Uncertain context, unverified | Dingo above dates of 800 ± 50BP (Beta-46318) and 1580 ± 70BP (Beta-46090) | David 1992:51 [76] | |
| Therreyererte | Central Australia | 1 | Excavated from alluvial fan deposits | Pseudo-burial | Dingo just below date of 400 ± 50BP | Smith 1988:288–289 [77] | Site contains cultural material but dingo appears to be natural burial |
| Wellington Square | Perth | 1 | Excavated from ground | Buried in ground | Possibly 19th or early 20th century | Anon. 1942 [78] | Associated with human burial |
| Tamboon Inlet | Gippsland | 1 | Excavated from occupational site | Buried in sand | | Anon. 1936 [79] | Associated with 2 human burials, occupational debris |
| Brutus Creek, Ilfracombe | Central Queensland | 1 | Excavated from surface-exposed burial pits | Buried in ground | | Anon. 1935 [80] | Associated with 2 human burials and occupational debris |
| Williamstown Racecourse, Melbourne | Melbourne | 1 | | Buried in ground | | Anon. 1915b [81] | Associated with skeleton, others previously found at site |
| Barwon Station | Northern NSW | 3 | Collected from rabbit warren | Buried in ground | Potentially post-contact due to provenance | Anon. 1948 [82] | Associated with human burial |

(*Continued*)

**Table 1.** (Continued)

| Site Name | Region | No. of burials | Context | Burial mode | Dating | Source | Comments |
|---|---|---|---|---|---|---|---|
| Native Creek, Meredith | Western Victoria | 2 | Collected from in or near rabbit warren | Buried in ground | Potentially post-contact due to provenance | Anon. 1915a [83] | Dingo skeletons had major cranial injuries, found lying at the feet of two human burials |
| Yilki, Encounter Bay | Lower Murray | 1 | | Buried in ground | | Anon. 1960 [84] | Associated with multiple human burials |
| Leaghur Station, Kerang | Northern Victoria | 1 | Excavated from burial mound | Buried in purpose-made burial mound | c. 1844 AD | Mathieson 1934 [85] | Associated with 5 human burials. Oral history identifies them as casualties of tribal conflict 1840s |
| Wandearah West | Spencer Gulf | 1 | | Buried in ground | | Anon. 1924 [86] | Associated with human burial |
| Madura Cave | Nullarbor | 3 | Excavated from occupational midden in cave | Unverified | Two specimens represented by postcrania directly dated with calibrated ranges 3080–3352 and 1932-2096BP. A third (I6-W) undated from more recent layer. | Balme et al. 2018; Milham and Thompson 1976 [87, 88] | Uncertainty over degree of skeletal representation, it is unclear which if any of these represent burials or whether they are food remains |
| C99, Cape Range | Gascoyne coastal | 1 | Excavated from rockshelter deposits | Buried in ground | Direct AMS date on molar, 760±20 (668–723 calBP) SANU-70629 | Przywolnik 2002:169 [89] | Dingo bones consisting of mandible and fragmented postcrania including phalanges recovered from below a date of ~2500 BP, but younger direct date indicates it is considerably more recent and so likely reached position through burial. Associated with occupation material |
| Rockshelter, name unspecified, Wardaman country | | 1 | | Burial–ochred bones | | Bruno David (Pers. Comms.) in Gunn et al. 2010:13 [73] | In association with human burials, rock art and occupation material. Whether buried in ground or stowed not specified |

happened to die within the rockshelter just before a flood event, but this is not clearly demonstrable nor is it suggested to have occurred outside one well-known site - Fromm's Landing/ Tungawa [51]. We have designated all of these as being unverified but likely of natural origins (Fig 1 and Table 1).

As previously noted by Brumm and Koungoulos, there is a slight tendency for dingoes in archaeological burials to be male, and a strong tendency to be either subadults [six months to one year] or young adults (≥1.5 years) [22]. The exception is a single individual from Mallacoota was estimated by Gollan to be more than four years old. This male was exceptional as it had suffered a severe traumatic injury to the upper back of the neck/caudal cranial area, which had healed completely by the time of its death. Gollan's interpretation was that the dingo had been reliant on the human inhabitants of Mallacoota for sustenance until this had occurred [34]. Dingoes are capable of breeding within their first year of life when environmental conditions are especially good, but in practice most reproduce closer to two years of age. Males are far more likely than females to reproduce before reaching two years old, and close to 100% that reach three years old have reproduced by this time [18].

Accurate understandings of the antiquity of individual dingo burials are limited. Apart from a few direct dates related here, interpretations of age have generally been made through supposed "association" with radiocarbon-dated archaeological charcoal or shell.

Unfortunately, in many cases the strength of "association" between the dingo and other dated material is dubious. Aside from where stratigraphy is unclear or disturbed, the action of ground burial via interment by nature inserts younger bones into older sediments/materials. Without careful, specific recording of the in situ position of dingo burials and a precise understanding of their surrounding stratigraphy and features, such interpretations are likely to be misleading. Unsurprisingly, direct dates have revealed some dingo burials to be younger than what their stratigraphic position might have originally suggested [52], or in other instances *older* than all nearby and ostensibly "associated" human skeletal material [53]. In all reported cases there is no reason to believe that the burials occur any earlier than the late Holocene.

The distribution of recorded archaeological burials in Fig 1 is uneven, with the vast majority being recorded from two areas of southeastern Australia: the lower Murray-Darling River region, and along the southern coast of eastern Australia. Interestingly, these are the two areas that Gollan [12, 34] suggested that tamed dingoes had been reliant on human provisioning and that they had undergone morphological changes perhaps as a result of isolated breeding pools. For the Kioloa dingo burial, Gollan suggested that observed reduced body size, dentition, and auditory bullae were evidence of restricted breeding [34]. When Gollan undertook this work, he relied on a date from charcoal above the burial to suggest an age dating to about 200 years before European settlement (320±180 BP; ANU-2342). We have now obtained a direct date on a calcaneus from this skeleton of 1092±21 (SANU-72536), which provides a 95.4% probability range of 937–1059 calBP using the SHCal calibration curve [54].

A third notable cluster of dingo burials is evident in northwestern Australia, covering approximately the region from Arnhem Land (NT) to the Dampier Peninsula (WA). Unlike those of the southeast, these are almost entirely "crevice" burials in which the skeletons are not interred but placed deep into crevices, clefts or on hard-to-reach ledges in the walls of caves, rockshelters or other rock formations (Fig 2A). This burial method was commonly used for people across much of northern Australia [44] and forms part of the category of "stowing" burials, distinguished from interment burials, by Oxenham and colleagues [90]. Both canid and human skeletons were sometimes wrapped in a paperbark "coffin". If disturbed by animals, human skulls might be retrieved by relatives and placed alone on a ledge [73, 74]. Isolated dingo skulls have been noted in similar placements in northern rockshelter ledges or crevices, suggesting they may have received similar "curated" treatment after their secondary burial [75].

Historically recorded crevice burials are secondary; removed to their final resting places after exposure during their primary aerial burial produced clean, dried bones. An example of a dingo/dog aerial 'burial' was photographed on Sunday Island, just off the north western Australian coast during the Swedish Scientific Expedition of 1910–1911 (Fig 2B) and published by Akerman [91]. This photo is almost identical to that published in Love originally in 1936 [39]. Although not in use at the time of the expedition, the bier created in the tree on Sunday Island contained remains of three canids and two further crania and other bones were found on the ground below. The proliferation of individuals at this site might suggest that the bones were not removed for secondary disposal in this area.

For burials of this nature, only direct dates on bone (or paperbark or other organic wrappings) can give an accurate idea of antiquity. As they are exposed to the elements and numerous agents of disturbance, it may be expected they would not last as long as interment burials, and as such those few that are known might have only been deposited recently before their discoveries. Two such canid burials from Jawoyn country have provided 18-20th century AD AMS dates which support this hypothesis [73, 74]. In the discovery of the first burial it was reported that the site was not one used for human burials, and that Jawoyn people consulted on the matter considered it an unusual occurrence [73]. In the second instance no human skeletal evidence was found, but walled structures on nearby ledges suggested that human burials

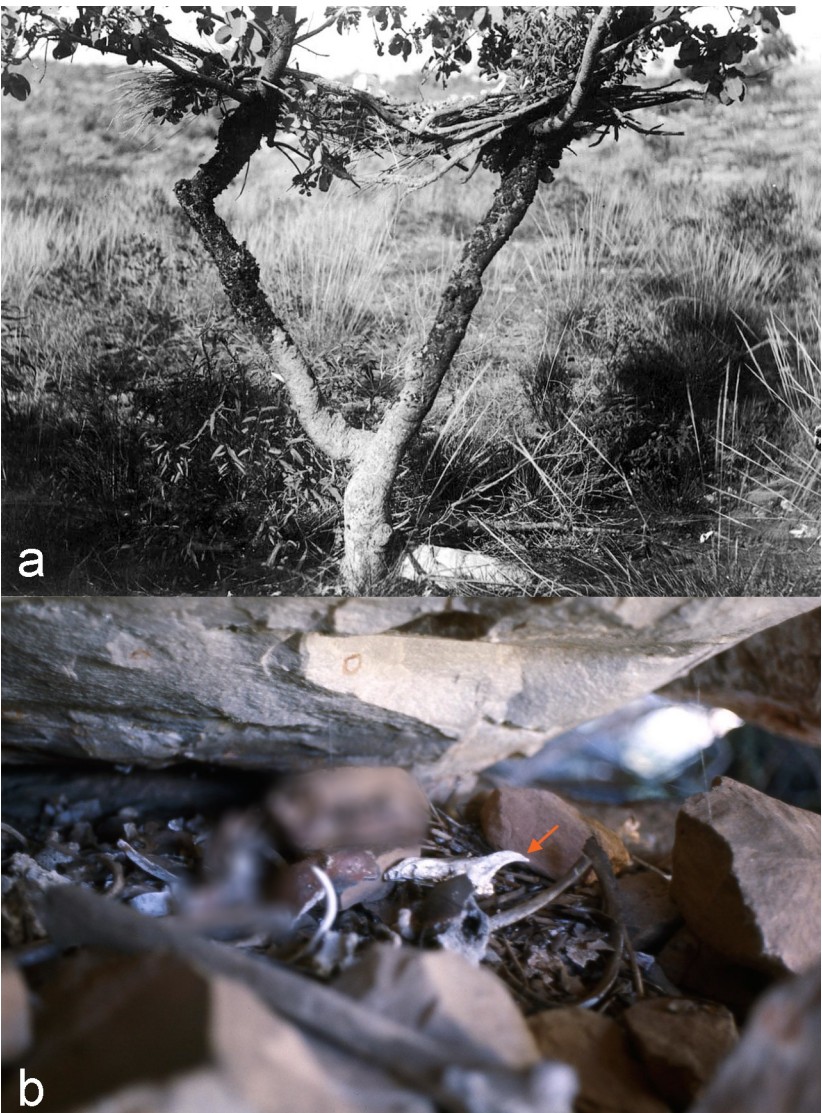

**Fig 2. Aerial and stowed canid burials in the Kimberley.** a: Burial tree with 2m high bier used for canid aerial burials on Sunday Island, western Kimberley photographed by the Swedish Scientific Expedition to Australia (1910–1911), Courtesy of the National Museum of Ethnography. Stockholm, Sweden. No. 0086a.067 [91]. b: Stowed canid burial recorded in a rockshelter crevice Adcock Gorge, central Kimberley, in physical association with human burial [elements censored]. Red arrow denotes canid left cranial fragment comprising incisive, maxilla and zygomatic bones. Photograph courtesy of K. Akerman.

had previously been located within the site. This second burial was also suspected on the basis of cranial morphology to be a domestic dog or dingo-dog hybrid [74].

It is important to note here that the discrepancy in burial modes–interment vs. stowed (or single-stage aerial)–between southeastern and northwestern Australia has created a geographic bias in the preservation of dingo/dog burials, and thus in impressions of their antiquity and frequency, which clearly favours the southeast. This might otherwise be taken (incorrectly) to reflect cultural differences in attitudes towards tame dingoes, or towards their inclusion in funerary rites, between southeastern and northwestern Australia, given that there are suggestions that dingoes were depicted more "negatively" in the lore and traditions of northwestern

Australia [43]. The discrepancy would appear to be a predominantly preservational issue, but it is also worth noting that many southeastern burials were discovered in the course of large-scale archaeological excavations relating to development projects within high-density metropolitan areas, so there is also probably a sampling bias at play [67].

In his 1982 study Gollan provides a cursory inspection of a single maxillary fragment of a dingo burial (1CU5/16) from the southeast coastal site of Curracurrang held in the Australian Museum's collections. The only comment made was that the upper third premolar was of diminutive size compared to its otherwise large (as normal for dingo) maxillary teeth [34]. However, there are several dingo burials at this site, which provide an opportunity to investigate further the evidence for the relationship and antiquity of humans and dingoes in this region. Here we present a primary osteological assessment of the individual (1CU5/16) inspected by Gollan, with further discussion of several additional finds of dingo identified during a recent systematic assessment of the 1CU5 faunal assemblage.

## Dingo burials at Curracurrang Rockshelter [1CU5]

Curracurrang Rockshelter (otherwise known as 1CU5) is a coastal Holocene site in the Royal National Park, approximately 30km south of Sydney, NSW, excavated between 1962 and 1966 by Megaw and team [61, 92, 93]. Studies of the site's material culture have made important contributions to understanding the regional chronology of backed artefacts, ground-edged axes, bone points, and shell fishhooks [94]. The site is also notable for its rare preservation of human remains in chronological association with a backed artefact industry [61]. However, of the faunal remains, only brief mentions of the presence of some larger taxa identifiable at the time of excavation have been published [93]. This included a mostly complete dingo burial from the wall of Square 16, Spit 2.

The stratigraphy and chronology of occupation of 1CU5 is complex, with both vertical and lateral inconsistencies in depositional ages apparent from the originally reported radiocarbon dates [61]. The initial transect's test pits (1962–63 season) were excavated on the basis of Megaw's aforementioned interpretation of depositional phases while the later excavations adopted arbitrary spit depths [usually six inches]. Megaw divided the sequence into three cultural periods. The uppermost was referred to as the "midden" layer by Megaw; so called because of the predominance of shell and fish bone with radiocarbon dates suggesting its deposition occurred between c.1000BP-1850AD [61]. This period forms the majority of the depth of cultural deposit within the rockshelter overhang, where it filled up within a large pit dug by the ancient occupants. It covers most of Squares 1–6 and parts of 8, 9 and 16 (Fig 3). The two remaining phases were labelled the "Bondaian" and the "Capertian" by Megaw, following McCarthy's Eastern Regional Sequence of regional technological development [95, 96]. The Bondaian middle phase contained a preponderance of Bondi points [a form of backed artefact] and has associated dates of c. 2500BP-1000AD. The earliest "Capertian" phase contained a primary flake and pebble-tool industry for which Megaw's radiocarbon dates span the period c.7500-2500BP [61]. A distinct lack of faunal remains was noted in both of these phases [93].

Based on these dates Megaw believed that the Midden and Bondaian deposits represented discrete, sequential occupational phases with little or no chronological overlap. He argued that the reason very few animal remains were found in the latter unit (and in the Capertian) was because of increasing soil acidity from the surface to base of the deposit presenting an unsuitable environment for bone preservation [61, 93]. However, he only obtained dates from the uppermost part of the midden, and from outside the rockshelter overhang on the slope where movement of sediment and materials possibly occurred. New AMS dates obtained on bone from the midden deposit within the rockshelter as part of our research indicate that the

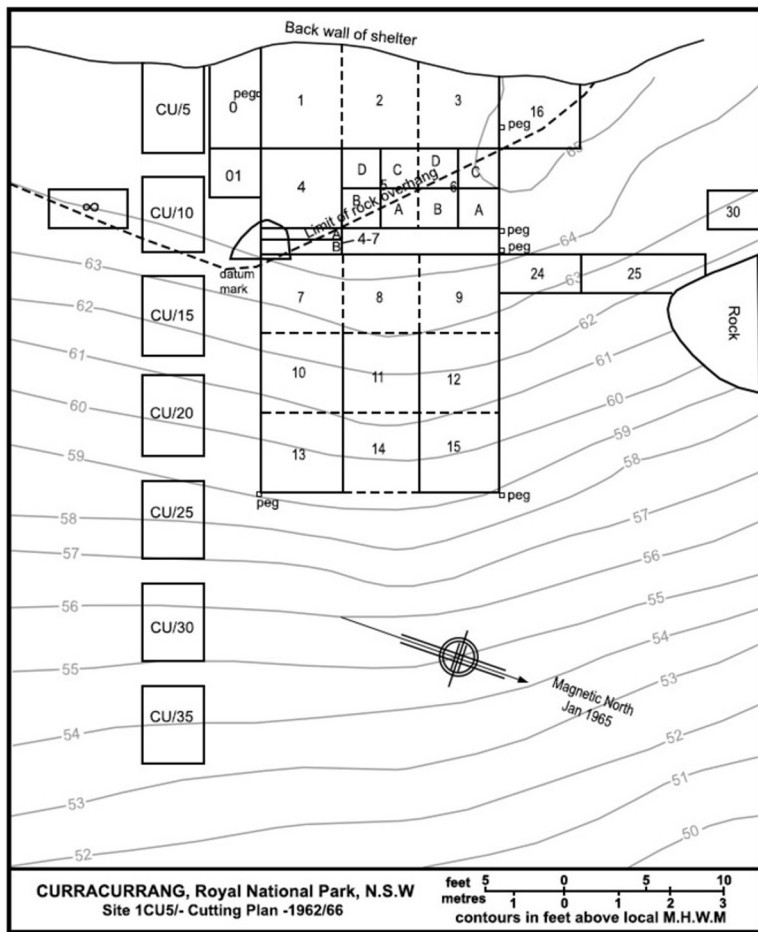

**Fig 3. 1CU5 site plan, redrawn from original by Branagan and Megaw [92].** Squares prefixed CU/- designate initial test pits from the 1962–63 field season transect.

midden unit began to form around or prior to 2500BP, and that much of it–recorded as "coarse midden" is actually contemporary with the upper part of Megaw's Bondaian phase (i.e. roughly 2500-2000BP). The uppermost part of the midden–recorded as 'loose banded midden'–would appear to have been deposited from after 1500BP until European contact.

Whilst a dedicated reassessment of 1CU5 site-formation is forthcoming, for the purposes of this paper it suffices to acknowledge that primary identifications of 1CU5 fauna found that dingo is distributed throughout the deposit, being present in nearly every square and in both the highest and lowest spits of the deepest-excavated squares. A minimum of 19, and possibly as many as 25 individual dingoes are represented in the material excavated from Curracurrang. We obtained direct AMS dates for six of these individuals, spanning almost the total depth of the deposit (Table 2). It is important to note here that the act of *burial*–inhumation below the surface–occurring during deposit formation may explain slight inversions in their ages. Nevertheless, dates from specimens from the top and bottom of the same or adjacent squares indicates that dingoes were present throughout the Bondaian and Midden phases (Table 2). The only dingo for which a near-complete skeleton was preserved is the individual from XU 16, also referred to here as the "main burial".

Gollan interpreted the 1CU5/16 main burial's stratigraphic position in the upper part of the "midden" as a sign of it being associated with the modern (c.1850AD) dates presented for a similar depth from different squares by Megaw [34, 61]. However, Square 16, at the northernmost edge of the area enclosed by the rockshelter overhang, was unaffected by the early Midden phase pit-excavation and consisted essentially entirely of intact Bondaian-phase midden deposit. AMS dates of samples taken from the mandible and femur (Table 2) correspond closely with Megaw's original date for Spit 4 from this Square (GaK-896; 2110 ± 90BP, or 2320–1830 calBP) which was from a hearth in the Bondaian unit [97]. No further information regarding its position or orientation within the deposit is available as no photographs, drawings or other recordings seem to have been made at the time of the burial's excavation.

Before Gollan's study, dentary fragments and most major appendicular elements of the main dingo burial had been identified and bagged separately from otherwise unsorted Square 16 fauna by Australian Museum research officer (1972–73) Peter Thompson [34]. Box and bag labels indicate the preliminary identification of much of the skeleton's postcranial bone was wallaby. That the cranial and postcranial elements were identified differently in this manner probably suggests that the skeleton was no longer articulated at the time of excavation, otherwise it should have been recognized as a single individual. Our assessment of the remaining faunal remains from Square 16 identified cranial fragments, vertebrae, manual/pedal elements, and a partial baculum belonging to this individual, confirming its sex as male. The assessment also made clear that some of the skeleton had been originally located within Spit 3. However, this has little significance as both spits derive from the same stratigraphic layer.

The condition of the skeleton provides the basis for interesting speculation as to the timing and manner of burial. Although mostly complete in terms of element representation, some of the appendicular elements have been damaged by carnivores, removing proximal ends and leaving pitting, punctures and gnaw marks on the remaining proximate surfaces. The damage is clearest in the region of the proximal humeri (Fig 4C and 4D), and from the scapulae, which have both been almost totally deleted. Most manual and pedal bones, the left distal femur, and the left distal tibia are missing and may have been consumed. Indirect gnawing may also be responsible for areas of lost bone on the proximal and distal regions of the right femur (Fig 4E). At this time (c.2000BP), the Tasmanian devil and thylacine had been locally extinct for over a millennium [98] and the only candidates for damage on this scale are other dingoes. Cannibalism is well-documented in dingoes and may be linked to locally higher densities of conspecifics [99, 100]. Observations of large mammal carcass processing by dingoes indicate the abdominal area is targeted first to gain access to entrails [101], followed by consumption of digits [102], with limbs also removed and de-fleshed [103].

Ostensibly, any cannibalism occurred *prior* to inhumation, otherwise the limb bones might reasonably be expected to have been found scattered throughout other squares or removed from the site altogether, based on the movement via transportation of appendicular elements observed in feeding trials of dingoes [103, 104]. Overall, the consumption process was clearly not well advanced in the case of 1CU5/16. The presence of limited carnivore damage suggests that the buried individual was not directly by the side of a human (or kept in captivity/confinement) at time of death, but perhaps had been roaming outside of the site and presumably was located by community members sometime after its death, before retrieval and burial of the carcass within the rockshelter.

The maxillary and mandibular dentition of the 1CU5/16 burial displays advanced levels of wear, concentrated on the molars and carnassial and decreasing in degree towards the incisors (Fig 4A). Wear is also advanced on both the anterior and posterior faces of the canine teeth. A similar distributional pattern was noted in other coastal NSW dingo burials at Kioloa and Murramurang. However, the particularly high degree of wear in 1CU5/16 is not present in any

**Table 2. AMS direct dates from Curracurrang dingoes.** Calibrated ranges produced in OxCal 4.4.4 using ShCal20 calibration curve (Bronk Ramsey 2021).

| Sample ID | Provenance/Specimen/Element | ¹⁴C age (years BP) | ± | Calibrated 95.4% probability range (years BP) |
|---|---|---|---|---|
| SANU-64831 | Square 16 Spit 2 [Main burial 1CU5/16, mandible] | 2105 | 22 | 2142–1996 |
| SANU-64837 | Square 16 Spit 2 [Main burial 1CU5/16, femur] | 2095 | 24 | 2090–1930 |
| SANU-64832 | Square 9 Spit 2, mandible | 807 | 20 | 726–667 |
| SANU-64838 | Square 9 Spit 2, mandible [duplicate] | 805 | 20 | 726–667 |
| SANU-64833 | Square 2 Spit 3, mandible | 2257 | 22 | 2331–2136 |
| SANU-64835 | Square 3 Spit 12, distal metapodial | 2088 | 21 | 2087–1929 |
| SANU-24477 | Square 3, Spit 5, tibia | 2049 | 26 | 2011–1890 |

other individuals (wild or tame, modern or ancient) photographed and studied by Gollan-including the Kioloa and Murramurang individuals that are much younger in individual age as indicated by incomplete alveolar and cranial suture closure [34]. Regardless, they together form a contrast to patterns described for wild dingoes in which attrition is more evenly distributed, with a high frequency of breakage amongst the premolars due to their role in the capture and processing of mammalian prey [34].

A degree of dietary differentiation from wild dingo is therefore inferred for the 1CU5/16 dingo, presumably involving greater proportions of hard and/or abrasive materials than featured in the typical wild dingo's diet. Dedicated analysis of the Curracurrang fauna is forthcoming, but our preliminary assessments find that carnivore damage is clearly evident on much of the mammal and fish bone. Frequent consumption of bone via scavenging of human meal discard or if human provisioning was limited to bone scraps as often noted historically

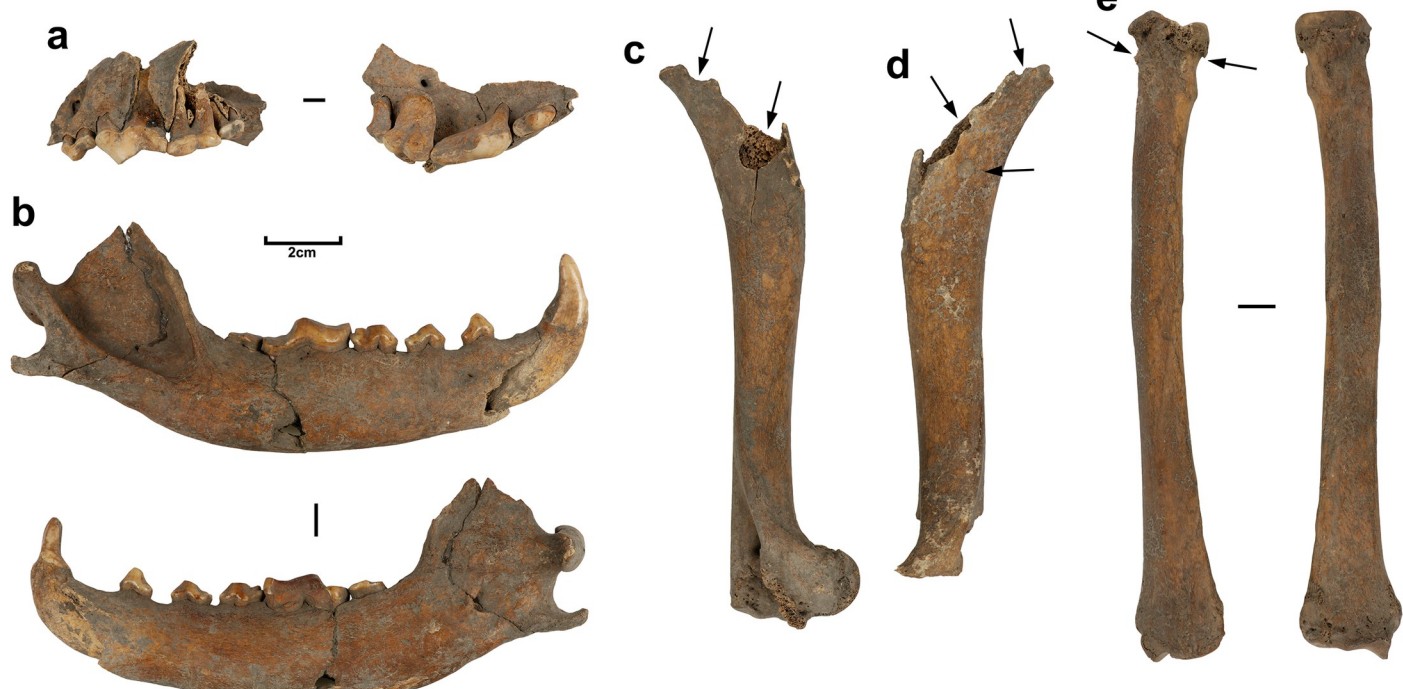

**Fig 4. Selected elements of 1CU5/16 dingo burial.** a) right maxillary fragment with occlusal view of teeth; b) lateral and medial view of right hemimandible; c) oblique view of right humerus; d) oblique view of left humerus; e) anterior and posterior view of left radius. Arrows on humeri denote carnivore damage; arrows on radius denote osteophytes.

[14]; might be responsible for elevated tooth attrition, as it is in wolves [105] and potentially incipient Palaeolithic dogs [106]. Higher levels of abrasive grit present in marine foods–seals and fishes–has been identified as an agent of similarly severe dental attrition in the New Zealand kurī [107, 108], which was a domesticated dog reliant on provisioning. Bones of marine fish were previously found in the stomach cavities of two other dingo burials on the southeast coast of Australia, suggesting they were a regular fixture in coastal tame dingo diets [34].

Whilst the degree of wear on the 1CU5/16 dingo's teeth may indicate an advanced individual age, it is possible that constant attrition from a bone-heavy diet could present a misleading impression in this regard. A further indication of adult age is given by the complete closure and fusion of epiphyseal sutures on all extant limb bones of 1CU5/16, such that the suture is in most places no longer discernible. The surviving articular surfaces of elements also exhibit some porosity and osteophytes, that are an indicator of osteoarthritis (Fig 4E). Osteoarthritis is generally increasingly likely with age in canids, but can also develop from an early age or following trauma [109]. Whilst there are no specific data on the epidemiology of osteoarthritis in dingoes, Gollan claimed that wild dingoes rarely survive long enough to develop osteoarthritis, with only one example of this disease noted in his survey of museum collections (modern and subfossil alike) [34].

Further pathological features - roughly circular lytic lesions–were observed on some vertebral spinous processes and endplates, on the pelvis, and in several articular and periarticular locations upon the limb bones (Fig 5). In the absence of evidence of bone infection or other microbial disease these lesions are most likely indicative of the growth of tumors [110, 111]; in this case, a malignant cancer such as multiple myeloma seems most likely. This plasma cell disorder is rare both in humans and dogs, but produces numerous conspicuous lytic lesions of relatively regular size in the bone surface and marrow, typically without associated new bone formation or peripheral bone reactions as seen in other potentially similar-presenting cancers [110, 112]. Multiple myeloma lesions are thus feasibly identifiable via differential diagnosis in archaeological specimens [112–114]. In dogs, it is significantly more frequently diagnosed in older animals and in non-neutered males [115], with an average age at diagnosis between 8 and 12 years old [116] but with cases recorded as young as 4 years old [117].

Multiple myeloma causes pain, and commonly lameness. If not treated with chemotherapy or modern drugs it generally leads to fatality within a short time after diagnosis [118]. The presence of numerous and fairly extensive lesions on the spine and limb joints in 1CU5/16 suggest that this dingo probably experienced limited mobility in the last period of its life and presumably was cared for by people until death. However, this is only a preliminary interpretation of 1CU5/16's observed pathological features, which require further investigation. Data on the incidence of cancer in dingoes are very limited, but it seems to occur less frequently within dingoes and other "ancient" East Asian varieties compared to European breed dogs, perhaps owing to a history of less intensive selection and inbreeding, and not overly large body-sizes [119]. Interestingly, it has been argued that the high prevalence of cancers in dogs in general is the product of human care-related increases in lifespan compared to non-domesticated wild *Canis* [120].

Without suggesting a specific age in years, the evidence from dental wear, complete long-bone fusion, and the presence of particular pathologies is sufficient to confirm that the 1CU5/16 individual was certainly well past the onset of sexual maturity and was of breeding age. Its relative completeness also allows for some commentary on the morphology and phenotype of this ancient southeastern coastal dingo. The reconstructed height at the shoulder of this individual following Harcourt's method [121] based on long-bone lengths is 49.89cm. This is ~9% below the overall mean for modern dingoes generated using the same method reported by Koungoulos, and ~5–10% below the mean for individual biogeographic populations, falling into the lowest end of any of their ranges [14]. Modern dingoes from the coastal areas of New

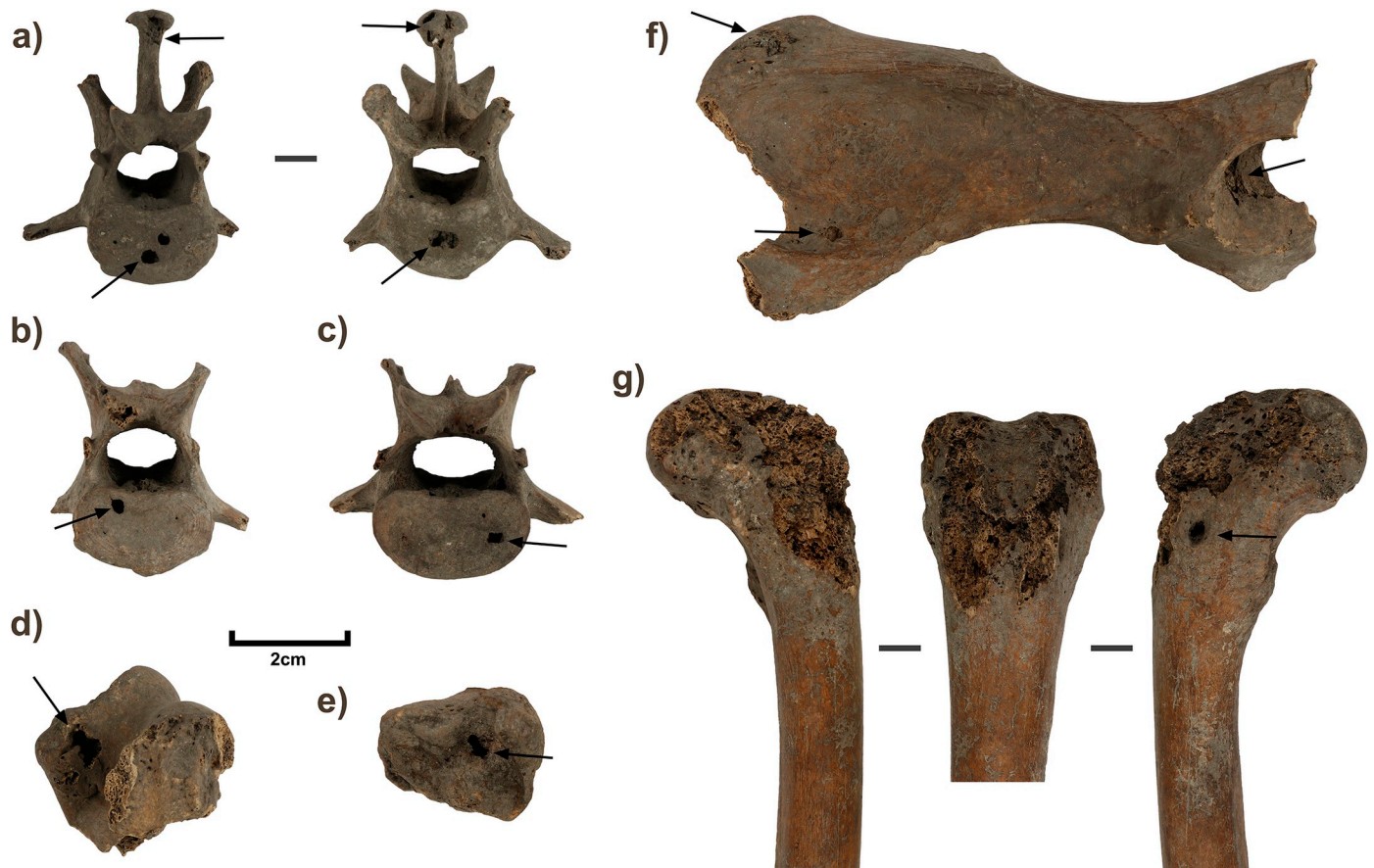

**Fig 5. Pathological features of 1CU5/16 dingo burial.** a) posterior and anterior view of lumbar vertebra; b) and c) posterior views of lumbar vertebrae; d) oblique view of distal left humerus; e) caudal view of distal right tibia; f) lateral view of partial left pelvis; g) medial, anterior and lateral views of distal right femur. Arrows denote lesions possibly associated with cancer.

South Wales south of Sydney, the most relevant population for direct comparison, range from 47.39–59.51cm tall with a median of 53.73cm and mean of 53.66 ± 3.56cm [14].

With regards to dentition, comparison of the maximum lengths of the 3rd and 4th (carnassial) maxillary premolars demonstrates that 1CU5/16 is well within the normal range for modern dingoes represented by six populations from southeastern, central, and northern Australia ($n$ = 243) (Fig 6). 1CU5/16's scores in both regards are towards the lower end of most modern ranges, as befits its slightly smaller overall size. Interestingly, most of the modern NSW south coast sample–i.e. that from the nearest geographic location and same biogeographic zone - falls into this region of the plot. There does not appear to be any proportional diminution of the third premolar compared to the carnassial as originally suggested by Gollan [34], rather, this was simply a somewhat small dingo compared to modern standards.

Slightly reduced overall size is a trait widely observed in archaeological dingoes studied by Gollan [34] and may be part of morphological changes related to ongoing selection and/or breeding isolation, though it may also be explained as recurrent consequence of insufficient nutrition during adolescence. Late 18th and 19th century European observers of camp dingoes often remarked on their poor nutritional state, and some specified their reduced size [14, 122]. Studies of dingo subfossil material from other parts of the continent also indicate a broad but modest increase in body size over the last ~1500 years [14]. Some of this likely occurred within

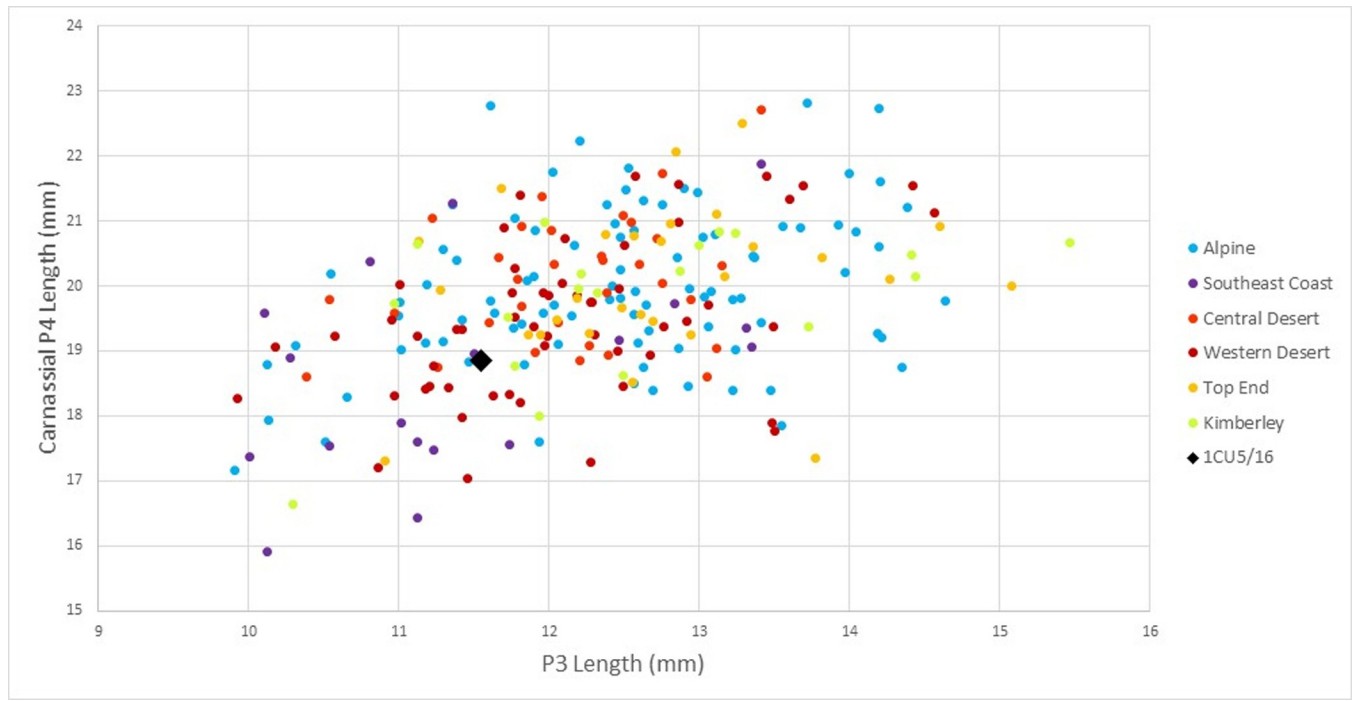

**Fig 6. Scatterplot comparing maximum length of maxillary P3 and P4 teeth in 1CU5/16 and 243 modern dingoes from six biogeographic populations.**

the last 200 years, as a result of admixture with large dog breeds and selection for larger bodies driven by poison-based control programs following European settlement [123]. Overall the morphological conformation of 1CU5/16 seems consistent with modern dingo from southern coastal New South Wales.

The remains of several other adult dingoes of varying personal ages were identified from the Curracurrang fauna. Most of these finds constitute single teeth or small clusters of permanent teeth, which are too numerous and insignificant to discuss in detail. In addition to these, there are several single finds of craniomandibular and postcranial elements, generally from non-adjoining Squares, which could not be matched to one another. We do not wish to infer much of their original depositional context as the limited information about their provenance and within Square contexts allows equal likelihood that they are disaggregated parts of disturbed and/or incompletely excavated burials, or alternatively (wild) dingoes consumed by people.

The partial left hemimandible of dingo of young adult age - permanent molar teeth fully erupted but exhibiting little to no wear–was identified from the Spit 2 of Square 9, outside the rockshelter overhang. Associated with the upper layers of the Midden period deposit of dense shell midden material, this returned a calibrated AMS range well within the last 1000 years (Table 3). No potentially matching skeletal elements were found in surrounding Squares, with the possible exception of a distal right radius from adjacent Square 24. However, this is more likely to match a proximal left radius and partial pelvis found in Square 25. Other isolated finds of adult dingo include mandibular and dental fragments from Spit 2 of Square 15; a distal 2nd left metacarpal from Spit 2 of Square 7; metapodials and phalanges accompanied by a complete atlas from disturbed fill on the surface of Square CU/10; and a distal left tibia from Spit 3 of Square 4-7a (a baulk between the within-overhang Squares 1–6, and exterior Squares 7–15).

Two more substantial specimens probably are also likely to represent burials given the range of skeletal elements represented, although neither are nearly as well-preserved nor

**Table 3. Identified remains of dingo pups from Curracurrang.**

| Excavation Context | Elements represented |
|---|---|
| CU/5D & CU/5M | Right hemimandible, cranial fragments (maxillary and caudal), various upper and lower deciduous teeth |
| CU/5D | Right hemimandible |
| CU/5M | Left radius and calcaneus; right scapula and ulna |
| Square 0 Spit 1 | Right hemimandible with deciduous premolar |
| Square 0 Spit 2 | Right ulna |
| Square 1 Spit 1 | Left hemimandible, femur, and pelvic ilium; right scapula |
| Square 1 Spit 2 | Lower left canine and carnassial teeth; left humerus; right femur |
| Square 1 Spit 2 | Left pelvic ischium and right femur |
| Square 1 Spit 3 | Left scapula; left pelvic ilium; left tibia |
| Square 1 Spit 6 | Cranial fragment (incisive) |
| Square 2 Spit 1 | Cranial fragments (temporal region), lower right canine and carnassial; left femur; indeterminate fifth metatarsal |
| Square 2 Spit 2 | Cranial fragment (left temporal region) |
| Square 2 Spit 3 | Left and right hemimandibles, cranial fragments [maxillary and caudal], upper and lower deciduous teeth; right pelvic ilium; left distal femur; right distal tibia |
| Square 2 Spit 4 | Cranial fragment (caudal), right hemimandible; left humerus; fragments of indeterminate radius, femur and tibia; proximal rib |
| Square 3 Spit 3 | Deciduous canine tooth; indeterminate metapodial |
| Square 4 Spit 3 | Right hemimandible with teeth |
| Square 4 Spit 5 | Right hemimandible |
| Square 5B Spit 2 | Proximal left humerus |
| Square 6B Surface | Fragment of pelvis and medial left humerus |

complete as 1CU5/16. This is probably because of their provenance in squares at the edges of the excavated portion of the site. It is likely that missing skeletal elements lay within unexcavated deposit beyond their boundaries. Given their fragmentation and provenance within dense shell and fish-bone midden material, it is unsurprising that these two individuals were not recognised as potential burials at the time nor that extensions were not performed to recover the rest of the skeletons.

Heavily fragmented pieces of the mandible, incisive dentary, associated permanent teeth, and first three cervical vertebrae of an adult dingo were recovered from Spit 5 of Square 16 – below the 1CU5/16 burial. Comparison of fragments of the coronoid process, the largest identifiable pieces remaining of this individual, suggest that it was somewhat smaller than 1CU5/16. Megaw's date from the spit above this places this individual in [at least] the range of 2320–1830 years calBP [61]. Indeed it is possible that the metapodial found at a similar depth in the adjacent Square 3, which provided a new AMS date within this range (SANU-64835), belongs to this individual. Aside from a similar external bone surface condition to the craniomandibular fragments, this metapodial fragment was the only piece of bone found at this depth of Square 3 and an intrusive origin seems likely.

A second probable burial of an elderly dingo was identified from the top of the midden unit (i.e. just below the modern topsoil surface) in Square CU/20 from the initial test excavations. This skeleton is in far poorer condition than the other and is represented by craniomandibular elements, partial humeri, and left radius, and one proximal phalanx. As these are all from the head and forequarter of the body, and because the respective square is an isolated one at the edge of the site, it is probable that other surviving elements remain in surrounding

unexcavated sediment. This individual was comparable in size to 1CU5/16 and is evident from the extreme wear of the recovered molars alone that it was also of advanced age (Fig 7).

Numerous very young dingoes [pups] were also identified from the Curracurrang excavations. A minimum number of seven were identified based recurring mandibular elements (Fig 8). They were found in squares almost exclusively within the overhang of the rockshelter, or in one case (Square 6B) approximately on its dripline. Based on their small size, limited development of overall anatomy [rounded, curved form of overall body; soft, short, and ill-defined condylar and angular processes] and the presence of deciduous teeth with no sign of adult teeth yet to erupt, these individuals were at least 17–21 days of age and no older than eight weeks [18, 124]. One specimen from the disturbed portion of pit CU/5 is extremely small and may represent a more recent neonate (Fig 8E).

Another from Spit 3 of Square 2 (Fig 8A), with matching bones found from the same Spit in adjoining Squares 1 and 3, returned the oldest AMS date for dingo from this site, with a calibrated range of 2331-2136BP (SANU-64833) (Table 2). This individual is the most complete of the juveniles from Curracurrang, with much of the appendicular skeleton represented. Coverage of skeletal representation in the other pups is lesser but nevertheless notable (Table 3). Some of the materials reported from different (but adjacent) XUs in the table could belong to the same individuals, but were either lying on the boundaries of the squares, or had disaggregated and become separate due to disturbance (see specimens from CU/5D and CU/5M) or post-depositional movement of midden materials.

We believe that the consistent postcranial representation of most of these individuals indicates they are more likely to represent deliberate burials, rather than meal discard. The latter possibility must be seriously considered as historical consumption of dingo pups specifically was widespread, although it seems to have been generally observed well to the west of the Sydney region and predominantly in northern and central Australia [15]. Preliminary assessment of the Curracurrang fauna has noted that small mammals of comparable size are predominantly represented by isolated hemimandibles, with postcrania being infrequent, and it is rare that >2 postcranial elements co-occurred within the same excavation unit. That many dingo pups are commonly represented by cranial in addition to postcranial elements from throughout the body is therefore a distinctive taphonomic quality for this site, and suggests their deposition was probably as complete and articulated carcasses. As such, we are inclined to believe that this occurred through human agency. If this is correct, the Curracurrang specimens mark the first known evidence of the burial of neonatal to very young juvenile dingoes that we are aware of–other instances of "young" dingo burial all seem to be older than six months of age [14, 34].

The presence of numerous pup remains also raises an interesting question about the reproductive and genetic histories of ancient tame or camp dingoes. It is currently impossible to tell whether the Curracurrang pups were born on-site or whether these were juveniles taken from nearby wild dens which died soon thereafter. Given the presence of dingoes in and past the age of sexual maturity in residence at the site, and in the absence of evidence to the contrary, there is no apparent reason to rule out in-camp reproduction. Historical evidence suggests high mortality rates amongst pups born to more recent camp dogs of Aboriginal settlements, some of which were deliberately actioned by people [125]. The evidence at Curracurrang therefore potentially marks an important departure from the colonial-era mode of dingo pup procurement from wild dens.

Importantly, the implied continuity between successive populations of camp dingo presents a scenario in which traits relating to anthropogenic selective pressures could accumulate, potentially eventuating in the phenotypic changes observed by Gollan in other NSW examples [34]. A robust comparison with a sample of wild-born dingoes of equivalent ages could potentially reveal morphological differences owing to putative genetic isolation as discussed above.

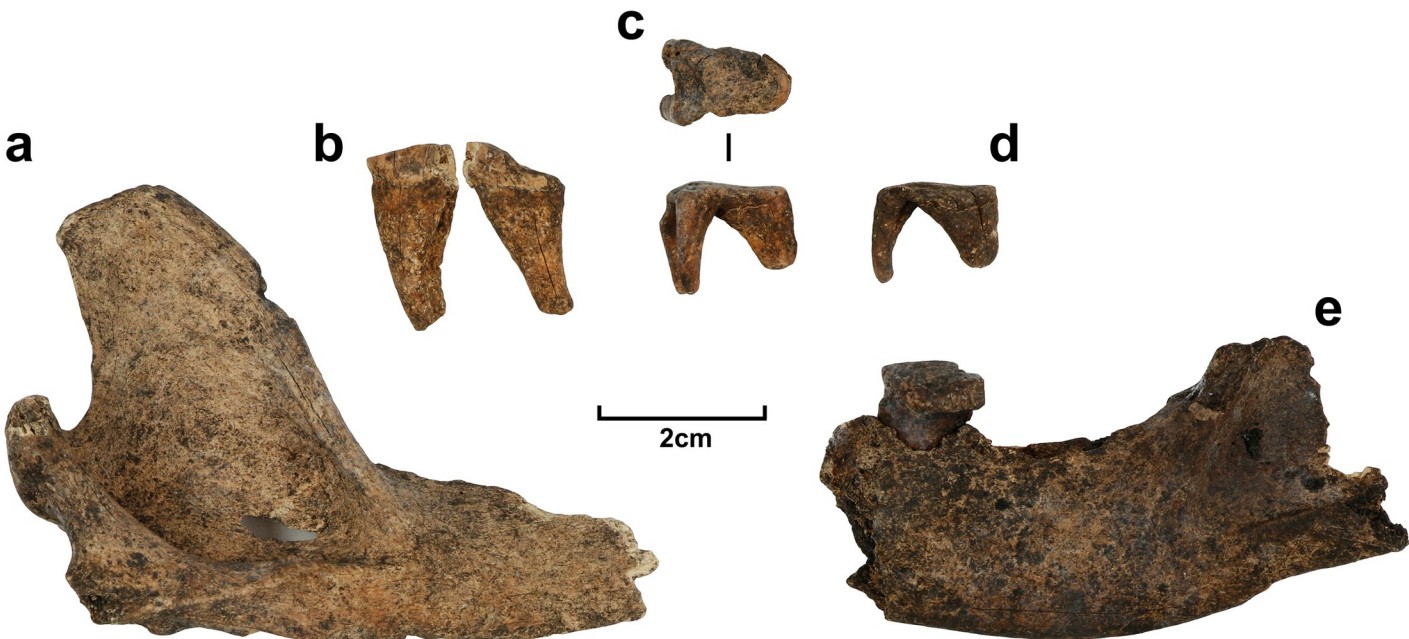

**Fig 7. Mandibular and dental elements of the second older adult dingo from the upper midden of Square CU/20.** a) lateral view of right hemimandible; b-d) views of carnassial and lower 2nd molar teeth; e) lateral view of left hemimandible.

Regrettably, no such collection is available at present. More importantly, it is entirely possible that morphological differences observed by Gollan and others are developmental or epigenetic in origin, and therefore may not be expressed in near-neonatal pups [14]. In any case, the distinction between wild-born and camp-born pups may not have been relevant to their human companions as it pertains to the question of their being given burial rites.

## Discussion

Dog burials, and the association of dog burials with human burials, have been recorded throughout the world wherever dogs travelled with people. The lineage of ancient East Asian dogs to which the dingo and its closest relatives belong are well-represented in burial contexts, from their origins in present-day China and through their vast networks of dispersal throughout mainland East Asia, Mainland Southeast Asia, Island Southeast Asia, and Oceania [126, 127]. Of particular interest to the issue of dingo burials are dog burials and remains from the regions nearest to northern Australia, all of which date from the Late Holocene. Interestingly, many of these cases seem to be dated very near to the oldest direct dates for dingo in Australia c. 3000–3300 calBP [5].

The earliest directly dated dog burial in regions near to northern Australia is found in the island of Timor. At the cave site Matja Kuru 2 at the far eastern end of Timor-Leste a dog burial was recovered and directly dated to 2867 ± 26 BP (2921–3075 calBP; Wk-34931) [128]. Isotopic and morphometric analyses indicate that this Timor dog was well-nourished and had a diet very similar to Pacific pig, suggesting that it lived its life in a sedentary agricultural community [128]. Ancient DNA analysis of this individual suggests it was not closely related to dingoes or extant ISEA dogs, but belonged to a branch of the broader East Asian dog family which expanded into the region likely alongside Austronesian migration [129]. A partial dog skeleton from Hoekgrot Cave in southern Java [130] is also very likely to be a burial. Although the excavation was poorly recorded, it was associated with dated fauna (2870–2492 calBP) and human

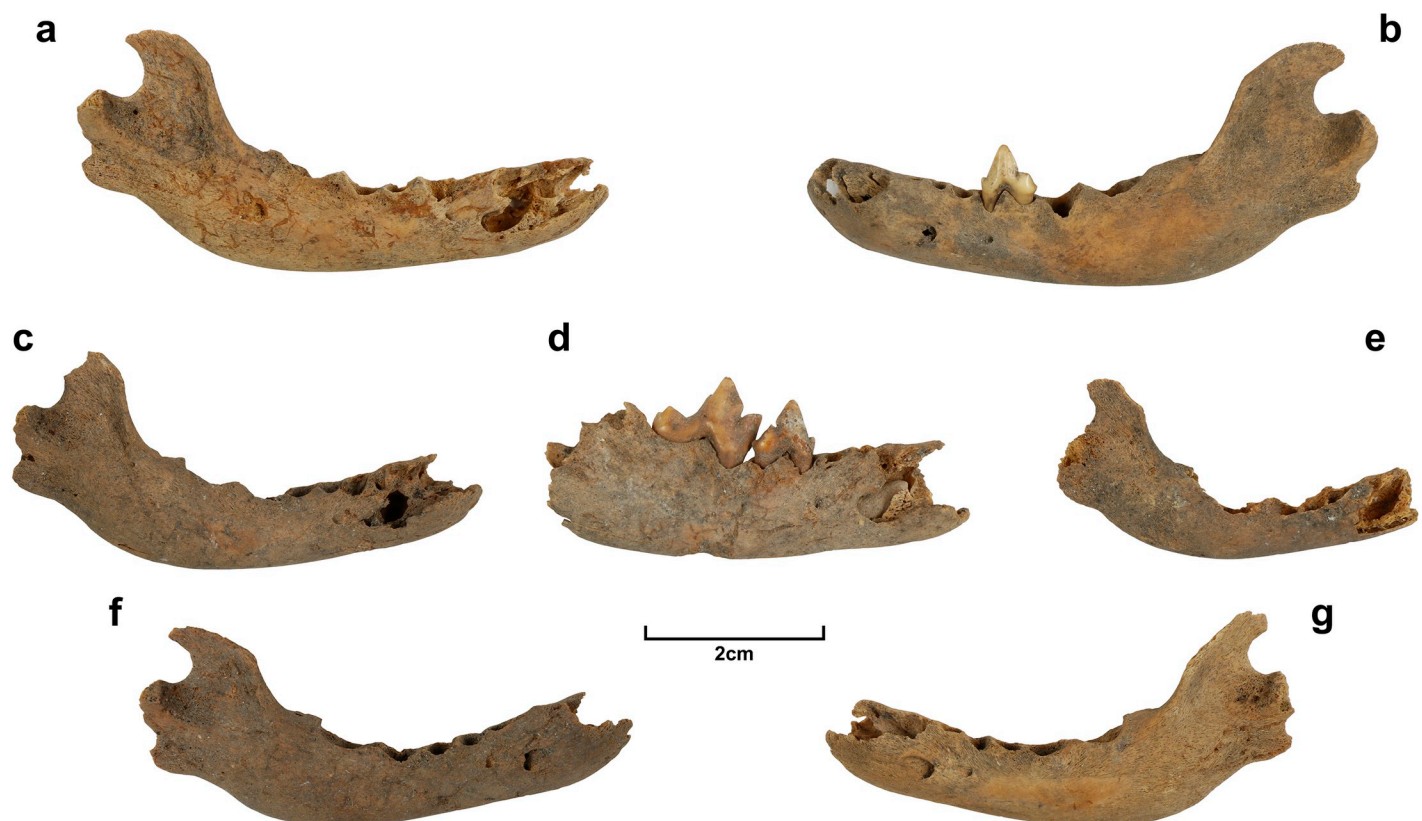

**Fig 8. Lateral views of juvenile dingo mandibles from Curracurrang.** a) Square 2, Spit 3 b) Square 0, Spit 1; c) Square 4, Spit 5; d) Square 4, Spit 3; e) Square CU/5, Disturbed fill; f) Square 2, Spit 4; g) Square 1, Spit 1.

remains (3578–3269 calBP). Fragmentary remains of dog are known from the Niah Caves in Borneo, dated by stratigraphic association to between 4820–4297 calBP and 3139–2756 calBP. There is no particular indication that these early Niah dogs were buried but later finds of dog [mandibles] in this site complex may have been ritually deposited (< 2500BP) [126, 131].

The earliest dogs in the Papua New Guinea region come from the Kamgot site in the Anir Islands, New Ireland in an Early Lapita layer and are dated by association to between c.3300-3000 calBP [132, 133]. The site of Talepakemalai in the Mussau Islands has also produced near-contemporary dog bones from the earliest Lapita phase between 3234–3089 calBP and 3155–3020 calBP [134], and new research on Brooker Island in the Massim group has found dog associated with dates 3220–3070 calBP [135]. All three sites' specimens are very poorly preserved and it is not clear whether they could constitute burials or are more likely discarded meal remnants.

There is a dearth of clearly-defined archaeological dog burials on mainland New Guinea, perhaps due to aerial burials being the method used to dispose of dogs, at least in the Highlands [136]. Definitive mainland dog burials appear c. 2000BP at the site of Taurama in southern coastal New Guinea. Here, a genetically-sequenced dog burial from was found to be part of a "Pacific" clade consisting exclusively of ancient dogs from Polynesia and the Torres Strait [129, 137]. Its immediate ancestors are therefore very likely to have arrived in New Guinea as part of external post-Lapita contacts with nearby Near Oceania. A poorly-preserved Late Lapita-associated dog mandible from the nearby site of Moiapu 1, dated to between 2573–2702 calBP

described by Manne and colleagues [132] is at present the oldest well-dated evidence for dog in mainland New Guinea, but this lone element has not been identified as a burial.

Whilst the incipient Lapita culture clearly possessed dogs it is unlikely they were of dietary significance like the pig and chicken [138]. This is because dog remains are very rare in early Lapita sites (i.e. in the island groups near New Guinea) and apparently totally absent in clearly-dated Lapita contexts outside of this region, suggesting they were not successfully established in new Lapita colonies outside of Near Oceania. Dog burials are reasonably common in parts of Remote Oceania settled by Lapita descendants where the species was successfully established [139, 140], but these are all much more recent cases (<2000BP). The longer-term establishment of dogs in Remote Oceania seems to have been facilitated by the introduction of a new type of dog belonging to the Pacific Clade that went on to found all pre-contact dog populations across Polynesia [141].

As such, at present the evidence for dog burial at the time of the earliest-dated dingo remains, c. 3000–3300 calBP, seems to be far more scant in the New Guinea-Melanesia context than it is in Mainland and nearby Island Southeast Asia [126]. However, it is important to note that there is a taphonomic factor in the mode of burial affecting survivorship rates, which may present a misleading impression of cultural practices of dog burial. The disposal of dogs in many interior and Highland locations of New Guinea in historical times occurred via aerial burial, analogous to that widely practiced across northern Australia [142–144] with inhumations perhaps more common in coastal areas [145] and Near Oceanic islands like New Britain [34]. Aerial burial was also the manner in which human bodies were disposed of, and like in Australia, switching to ground inhumation for both humans and dogs probably followed exposure to missionary activity [146]. Hence, dog "burial" in New Guinea could have a longer antiquity than is attested by surviving burials in the Lapita or Austronesian coastal contact zone like at Taurama.

There has been much discussion about the extent to which dingoes can be regarded as a 'domestic dog' or whether they should be regarded as a separate species. Those who advocate for a separate species status for dingoes [4] argue that they do not fulfil the criteria of domestication, which is traditionally understood to entail human-influenced selective breeding resulting in biological changes, intergenerational continuity and eventual dependence on humans. Those who argue for identification of dingoes as true domestic dog *Canis familiaris* [1, 2] interpret "domestication" in the more colloquial sense, which simply refers to recurring associations of humans and tamed animals. However, this looser definition might also suggest that various marsupials, lizards and birds captured and tamed by Aboriginal people were "domesticated", and there is no argument to assign these animals a separate species status [147].

Although arguments about the taxonomic status of the dingo also involve lengthy comparison of their morphological, genetic, and behavioural characters with those of domestic dogs, these are ultimately used to inform opinions on the dingo's domesticity or lack thereof as the key determinant of taxonomic identity. In many instances, however, interpretations of the same basic data vary considerably in whether they signal a wild or domestic identity. For instance, dingoes are argued to both differ in skeletal morphology from the typical domestic dog in ways that reflect natural rather than artificial or anthropogenic selection [4], but it is also recognised that dingo cranial morphology still falls within the large spectrum represented by domestic dogs, and is on the whole closer to certain dog breeds or landraces than to wolves or other wild canids [1, 2, 148]. Similarly, dingoes are genetically distanced from domestic dogs [149, 150] and hold fewer copies of starch-digestion alleles enriched in most domestic dogs [151]; yet these could be the result of genetic drift and natural selection during millennia of isolation and feral or wild-living lifestyles in Australia. Newer and more nuanced perspectives on what constitutes domestication aside from major phenotypic changes have not yet

been carefully considered in the case of the dingo, though it has become evident that such discussion is warranted [152].

Our study of the Curracurrang remains provides a timely opportunity to reassess this matter by making direct reference to archaeological material, rather than relying on inferences from the behaviour of recent or historical dingoes. With regards to the traditional (biological) perspective, the potential of phenotypic insights into of the domestication of dingoes at Curracurrang is limited by the fragmentary nature of the most complete individual skeletons. The ancient dingo 1CU5/16 bore a slightly smaller body and dentition compared to the majority of modern dingoes; reduction of body and tooth size is a widely-held marker of mammalian domestication [153, 154]. However, these dimensions appeared to be entirely consistent with proportions exhibited in the population found recently in the same geographic area (the south coast of New South Wales). Interestingly, the modern south coast population's reduced dental proportions have been prior interpreted as the result of heightened admixture from European dogs [155].

The appearance of these traits in a pre-contact individual suggests these traits could be attributed to natural regional variation, although it is possible that the modern instances are residual legacies of domestication-related processes which previously occurred in the region. With no natural (palaeontological) reference specimens from the same region of suitable antiquity to compare to 1CU5/16, there is an inability to differentiate between the outcomes of anthropogenic processes and adaptive localised variation. Equifinality of morphological outcomes resulting from very different selective processes is a major issue identified in archaeological investigations of the early stages of mammal domestication [156].

Turning to non-morphological assessments of domestication, Hulme-Beaman and colleagues have developed definitions and classifications of the diversity of commensal relationships between people and organisms living in the domestic environment (including anthropogenically-modified surrounds) [157]. These categories vary according to the degree of benefit derived by the human and commensal taxa, and correspondingly the dependency each create with one another. "Domestic" taxa are defined as those which humans "actively buffer from external selective pressures, enhancing survival, and breeding" [157]. Koungoulos has previously argued that postulated, but undemonstrated, pre-contact lifeways of tame dingoes which continued to reside in or immediately around Aboriginal camp throughout their lives [15], as envisaged by Brumm [21, 22] best fit the category termed "synanthropic commensalism". This refers to taxa that persistently inhabit domestic environments, but do not derive particular benefit from nor develop dependencies on their interactions with humans/ anthropogenic environs [157].

The most basic, conservative interpretation of archaeological dingo remains from Curracurrang - the enduring presence of adults in the camp - clearly fits the synanthropic commensalism category. However, we would also argue that the likely reliance of said dingoes on anthropogenic sources of food, their reaching advanced ages rarely seen in wild dingoes, their rearing of litters in the camp, and care by humans during illness are indicative of altered selective pressures and arguably enhanced survival and/or breeding capabilities. As such, there is some evidence from Curracurrang for domestication as defined by Hulme-Beaman et al. [157], though evidently not of its advanced stages in which humans directly control selective pressures.

More recent characterisations of domestication also emphasise its nature as an ongoing process, at variable paces and subject to interruption and reversal, rather than exclusively a single rapid event fixed in time with a permanent outcome. They emphasise that domestication occurs at variable paces. Bogaard and colleagues envisage a broad definition, which covers relationships "not necessarily initiated or closely monitored by humans" but which "persist of myriad reasons" for "mutual benefits" [158]. At Curracurrang, there is certainly evidence of persistent relationships between humans and dingoes, which were somewhat beneficial to the

dingoes in terms of the availability of food, and perhaps an environment in which pups could be safely born. Benefits to humans are not obviously visible at this site, but feasibly include the historically-recognised abilities of dingoes as hunting aids and for security [19, 26] aside from the desirable social and personal "benefit" of their companionship, especially as juveniles [15].

Critically, Bogaard questions the relevance of phenotypic syndromes as requisite hallmarks of domestication, and raise the common occurrence of hybridisation and backcrossing with wild ancestors in taxa under "domestication", which may well obscure boundaries between wild and domesticated individuals [158]. Therefore, bioarchaeological evidence for rapid change in populations in the process of domestication may not be forthcoming, thwarting simple assignments of specimens into firm wild or domesticated categories. Under their understanding of domestication [158], the lack of notable differences in measurable traits between the ancient dingo 1CU5/16 and modern dingoes may be attributed to the likelihood that dingoes raised in the camp at Curracurrang bred with wild conspecifics in addition to amongst themselves. But it may simply be that the potential domestication captured in the archaeological record at Curracurrang occurred at a slow pace, and did not result in marked biological changes during the lifetime of 1CU5/16.

Losey's recent evaluation of domestication also highlights its nature as a multifaceted, ongoing process without a discrete "threshold" [159]. Using the example of dogs, he questions the generalised use of human selection for friendliness and following phenotypic changes as the sole indicator of a "completed" domestication. Although accepting such scenarios as the potential beginnings of domestication, Losey instead emphasises human efforts in care and taming [socialisation], which facilitates living and working with animals in the domestic environment as an integral component of ongoing, multi-generational domestication [159]. This is required in *every* generation to sustain domestication, not only the first. Feral animals whose ancestors may have undergone biological selection in the early stage of domestication but have since returned to the wild cannot be interacted with in the same manner.

Camp-residing dingoes at Curracurrang occur within a timeframe of a few to several centuries (allowing for undated individuals from the more recent uppermost layers), and as such we interpret them as evidence of people's ongoing efforts to repeatedly maintain domestic relations with their dingoes as outlined by Losey [159]. This taming appears to have been very successful regardless of whether pups were born in the site or obtained from nearby dens belonging to the same breeding population in the manner outlined by Brumm [21]. There is no evidence to suggest that Curracurrang dingoes were ever forced to remain in the camp through breaking or binding their limbs as puppies, as has been recorded ethnographically from other parts of the continent [14]. Hence, it can only be assumed that upon reaching maturity the adults were suitably motivated to stay by food opportunities, and/or the strength of personal relationships with the people and families that tamed them [21, 22].

Finally, the act of animal burial itself has been proposed as an indication of domestication. Shipman [152], in outlining the aspects of dingo lifeways that align with a domesticated status, cites Morey's contention that the interment of canids is "one of the strongest archaeological criteria that signifies their domestication" [160]. Morey's argument is that burial most clearly signifies the social importance with which people regarded canids that they deemed necessary to bury, often in the manner of people and often alongside people, regardless of the biological status of the animal in question. In other words, burial directly reflects the fact that people had integrated canids into their domestic environments, both in the physical/environmental and social sense, marking them as "about as close to being considered a person as a non-human animal can be" [160].

Of course, this relies on buried animals having shared some relationship with humans prior to death, and not being purely wild individuals slain and buried in other circumstances (e.g.

sacrifice) [161]. At Curracurrang, the primary evidence for relationships between people and the buried dingoes are the presence of older animals with evidence of diets altered by anthropogenic resources. Even though we cannot rule out that pup skeletons may represent wild den-collected and not camp-born individuals, the decision to inter them nevertheless indicates the same conceptual inclusion into the domestic environment, and is reflective of the love shown by historical Aboriginal people [particularly women] for dingo puppies [15, 19, 21]. Not all dingo remains from Curracurrang represent burials or tame animals, however, in particular the several isolated elements which may be meal remnants.

It is interesting that both dingoes and people were buried within the rockshelter, and probably during the same occupational phase. Megaw reported a human burial associated with charcoal dated to 2360 ± 90BP (GaK-896) [61]. The lower end of its calibrated range (2705–2155 calBP) overlaps with the dates and ranges from several of the dingoes dated in this study, particularly SANU-64833. This young adult was excavated from the "Bondaian" unit, and so initially not considered to have any chronological overlap with dingo remains. However, in the midden unit that produced nearly all of the dingo remains, Megaw also reported burials of at least two ~3yo infants, two juveniles, one adult, and an isolated skull of a ~30yo woman [61]. Unfortunately, their precise stratigraphic locations were not reported, and because this unit was deposited over ~2000 years, it cannot be determined whether any of the human and canid skeletons should be considered co-burials. Regardless, it is clear that dingoes were buried in the same immediate space that Aboriginal people were.

The interpretation of dingo remains from Curracurrang therefore meets the basic criteria of presently prominent and emerging definitions of domestication [157–160] although the evidence for traditional or biological domestication is inconclusive, due in part to fragmentation and low sample size. However, there is a caveat when relating these findings to the dingo's taxonomic classification that should be acknowledged. Most dingoes at the time of earliest European contact were wild animals that did not rely on or even interact with humans to maintain stable, independent populations, as is true for the vast majority of dingoes alive today. The common presence of dingoes in naturally-accumulated palaeontological deposits such as pitfall traps suggests this was also the case for the three millennia for which there is firm radiocarbon evidence for canid presence in Australia [6, 14, 34]

Evidence for domestication processes affecting dingoes living in the company of pre-contact Aboriginal people is thus, at present, geographically restricted, and not representative of the situation for dingoes on the whole, as a distinct taxon or evolutionarily significant unit. Accordingly, our results do not provide a strong attestation to the taxonomic status of the dingo insofar as this is related to domestication occurring within Australia. In any case, the changing nature of anthropological discourse on what precisely defines domestication is liable to cause issues to arise in even the basic classification of "domesticated" species under binomial nomenclature, since it increasingly applies to taxa or populations which may bear no discernible biological difference to their wild ancestors.

## Conclusions

Dingo burial is a tradition that began from the time of, or within about 1,000 years of, the dingo's arrival in Australia, and appears to have been a widespread cultural practice. Both written and Indigenous oral historical records indicate that the practice was widely distributed throughout Australia at the time of European invasion. Not all camp dingoes were given burial rites, but in all areas in which the burials are recorded, the process and methods of disposal are identical or almost identical to those associated with human rites in the same area. This reflects the close bond between people and dingoes, and their 'almost human' status. Differences in

climatic or geochemical conditions conducive to the preservation of organic materials, combined with variation in the manner of burial has created a geographic discrepancy between the archaeological visibility of dingo burials favouring southeastern Australia. Both male and females, and individuals of all ages are represented in archaeological dingo burials. Dingo burials frequently or even mostly occur in sites where people are also buried. Spatial arrangements suggest some dingo burials from southeastern Australia could be co-burials with particular individual people, but possible cases remain to be verified with radiocarbon dating.

The rockshelter site Curracurrang, in coastal southeastern Australia, provides an opportunity to explore the occurrence and nature of dingo burial through the numerous dingo skeletons. Here, around 2000 years BP, tame dingoes continued to inhabit the camp into old age, consumed food most likely derived from humans, and potentially reproduced within the camp or its immediate surroundings. These findings suggest that people at Curracurrang maintained lifetime associations with individual dingoes. As such they are largely contrary to expectations of pre-contact tame dingo behaviour developed from contact-era ethnographies, which are mostly derived from arid interior and northwestern Australia. The tamed dingoes of Curracurrang meet many of the criteria of domestication, particularly when considering newer perspectives that emphasise enduring social relationships between humans and animals over major biological changes. However, this does not necessarily comment on the dingo's taxonomic status as a whole.

Dingo burials can be envisaged as the Australian representations of widespread Asia-Pacific practices of dog burial. Dog burials appear to be practised from the earliest times dogs arrive in the region, although at present the evidence for dog burials close to the time of the earliest directly dated dingoes is clearest in present-day Indonesia rather than in New Guinea or other parts of Melanesia. Dingo burials are at present an understudied resource, but their characterisation and interpretation situates Australia within a global phenomenon demonstrating the close bonds between people and their canids throughout antiquity, including the very earliest unambiguously identified domesticated dogs [109, 160, 162–165].

## Acknowledgments

We thank the La Perouse Local Land Council and to Batemans Bay Local Aboriginal Land Council for permission to carry out the work. Staff at the Australian Museum are thanked for helping us with access to the collections, and museum study permits: Allison Dejanovic, Mariko Smith, Rebecca Jones, Dale Higginson and Niamh Formosa. All necessary permits were obtained for the described study, which complied with all relevant regulations. Aboriginal Heritage Impact Permits (AHIP) for the destructive sampling of dingo bones from Curracurrang and Kioloa were obtained from the Office of Environment and Heritage, NSW (AHIP numbers 4353 and 4935). We are also grateful to Luc Janssens and Dennis Lawler for their interpretation and opinions on the pathological features of the 1CU5/16 dingo burial.

## Author Contributions

**Conceptualization:** Loukas George Koungoulos, Jane Balme, Sue O'Connor.

**Formal analysis:** Loukas George Koungoulos.

**Funding acquisition:** Jane Balme, Sue O'Connor.

**Investigation:** Loukas George Koungoulos, Jane Balme, Sue O'Connor.

**Methodology:** Loukas George Koungoulos, Jane Balme, Sue O'Connor.

**Project administration:** Loukas George Koungoulos, Jane Balme, Sue O'Connor.

**Resources:** Jane Balme, Sue O'Connor.

**Supervision:** Jane Balme, Sue O'Connor.

**Validation:** Loukas George Koungoulos, Jane Balme, Sue O'Connor.

**Visualization:** Loukas George Koungoulos, Jane Balme, Sue O'Connor.

**Writing – original draft:** Loukas George Koungoulos, Jane Balme, Sue O'Connor.

**Writing – review & editing:** Loukas George Koungoulos, Jane Balme, Sue O'Connor.

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
