## [Decision Letter · Decision Letter 0]

2 May 2023

PONE-D-23-07642Dingoes, companions in life and death: the significance of archaeological canid burial practices in AustraliaPLOS ONE

Dear Dr. Koungoulos,

Thank you for submitting your manuscript to PLOS ONE. After careful consideration, we feel that it has merit but does not fully meet PLOS ONE’s publication criteria as it currently stands. Therefore, we invite you to submit a revised version of the manuscript that addresses the points raised during the review process.

The reviewers judged this as a fine paper, and I agree with their assessment. Reviewer 2 has some minor recommendations for improvement. This includes citation of additional literature, which should be considered.  Please submit your revised manuscript by Jun 16 2023 11:59PM. If you will need more time than this to complete your revisions, please reply to this message or contact the journal office at plosone@plos.org. Please include the following items when submitting your revised manuscript:A rebuttal letter that responds to each point raised by the academic editor and reviewer(s). You should upload this letter as a separate file labeled 'Response to Reviewers'.A marked-up copy of your manuscript that highlights changes made to the original version. You should upload this as a separate file labeled 'Revised Manuscript with Track Changes'.An unmarked version of your revised paper without tracked changes. You should upload this as a separate file labeled 'Manuscript'.If applicable, we recommend that you deposit your laboratory protocols in protocols.io to enhance the reproducibility of your results. Protocols.io assigns your protocol its own identifier (DOI) so that it can be cited independently in the future. For instructions see: https://journals.plos.org/plosone/s/submission-guidelines#loc-laboratory-protocols. Additionally, PLOS ONE offers an option for publishing peer-reviewed Lab Protocol articles, which describe protocols hosted on protocols.io. Read more information on sharing protocols at https://plos.org/protocols?utm_medium=editorial-email&utm_source=authorletters&utm_campaign=protocols.

We look forward to receiving your revised manuscript.

Kind regards,

Michael D. Petraglia, Ph.D.

Academic Editor

PLOS ONE

Journal Requirements:

2. In your manuscript, please provide additional information regarding the specimens used in your study. Ensure that you have reported human remain specimen numbers and complete repository information, including museum name and geographic location. 

For more information on PLOS ONE's requirements for paleontology and archeology research, see https://journals.plos.org/plosone/s/submission-guidelines#loc-paleontology-and-archaeology-research

Please review your reference list to ensure that it is complete and correct. If you have cited papers that have been retracted, please include the rationale for doing so in the manuscript text, or remove these references and replace them with relevant current references. Any changes to the reference list should be mentioned in the rebuttal letter that accompanies your revised manuscript. If you need to cite a retracted article, indicate the article’s retracted status in the References list and also include a citation and full reference for the retraction notice

Reviewers' comments:

Reviewer's Responses to Questions

**Comments to the Author**

1. Is the manuscript technically sound, and do the data support the conclusions?

Reviewer #1: Yes

Reviewer #2: Yes

2. Has the statistical analysis been performed appropriately and rigorously? 

Reviewer #1: Yes

Reviewer #2: N/A

3. Have the authors made all data underlying the findings in their manuscript fully available?

Reviewer #1: Yes

Reviewer #2: Yes

4. Is the manuscript presented in an intelligible fashion and written in standard English?

Reviewer #1: Yes

Reviewer #2: Yes

5. Review Comments to the Author

Reviewer #1: I note a few typographical errors. Page 3, lines 51-53: A number of untamed wild animals have been transported by boat, so I do not endorse the idea that transportation invariably indicates domestication. Could dingos have been transported when very young? Page 6. line 132; 'purused" substituted for "pursued". page 11 line 293: "bird" substituted for "birds" ; page 14, line 313: language flow is garbled. Suggest deleting ""such".

The manuscript is an important, even-handed, sensible, and timely review of the behavior of indigenes of Australia and their feelings about dingoes from early accounts. Such data (diaries, letters, written accounts) are unlikely to be available outside of Australia or Oceania but are extremely pertinent to ongoing debates about whether dingoes are domesticated, whether or not dingoes are the same species as domesticated dogs, how dingoes might have spread throughout Australia and Near Oceania, and what circumstances might or might not have influenced treatment of dingoes possibly leading to domestication. In addition, the authors review archaeological or paleontological finds of dingo remains, which give strong support to assertions that dingoes were often buried or treated after death in a manner very similar to that observed for the dsposal of human remains -- and sometimes within areas used for human disposal. The authors are to be congratulated on seeking records of Aboriginal treatment of dingoes before modification or contamination by colonials and their domesticated animals.

This paper unquestionably deserves publication and attention. It is outstanding.

Reviewer #2: The dingo was introduced to Australia in the late Holocene and is primarily a wild animal. However, some dingoes were raised by Aboriginal people and lived in their camps until sexual maturity, after which they returned to the wild to reproduce. Some of these dingoes died while in the company of people and were buried in occupation sites, a practice that has received little research. Dingoes were buried in the same manner as Aboriginal community members, sometimes alongside people, and this practice likely occurred from the time of their introduction until soon after European colonization. The authors present a case-study of dingo burials from Curracurrang Rockshelter in New South Wales which provides insights into the lives of ancient tame dingoes, suggesting that domestication and genetic continuity may have occurred prior to European contact.

The ecological and historical impacts of dingoes, their taxonomic identity and their relationships with Australian Aboriginal people, has presented a lot of debate in the scientific literature. It does not help those perceptions of dingoes is mixed within the community, with some recognizing their unique ecological and cultural role, while others just regard them as pests. This manuscript is a very detailed description of dingo burials in Australia and presents some evidence that suggests domestication. The authors, however, keep a cautious tone, which is very welcome to see in the various debates about dingoes.

The writing, descriptions and figures/tables are all well presented and very detailed. What is missing is some of the subtleties of the debate when they are mentioned, and the missing of some key literature. These are mentioned in the minor comments below.

Minor Comments

Line 14: The Jackson papers argued for Canis familiaris not Canis familiaris dingo. It will be best to cite Smith et al. and Crowther et al .and acknowledge the differences of opinions in the literature. The recent genetic work of Kylie Cairns, demonstrating the differences between dingoes and dogs, is worth mentioning here.

Line 51: There are good references about the history of the introduction of cats and foxes into Australia including (Abbott 2002; Abbott 2011; Abbott, Peacock et al. 2014; Fairfax 2019).

Line 88: The arguments of Bill Ballard and coauthors regarding domestication and the dingo are also work exploring (Ballard and Wilson 2019; Field, Yadav et al. 2022).

Lines 525-542. These arguments for the taxonomic recognition are one of the arguments, and were certainly explored by Ballard and Wilson (2019), but they are not the sole argument of either camp in the dingo taxonomy debate. This often comes down to whether domesticated forms also deserve separate names to their wild ancestors, particularly where they wild ancestors’ direct populations may be extinct. This is beyond the scope of this paper, but certainly worth discussion in another forum. It should be briefly mentioned about how much the argument is about domestication, and how much about genetic (see the work of Kylie Cairns), morphological, behavioral and ecological differences.

In addition, there appears evidence here of phenotypic divergence due to secondary domestication of a population of wild dingoes. Although exciting and significant findings, these may actually not add much to the debate on dingo taxonomy, as mentioned by the authors in Lines 721-722. It may be more relevant in the literature about domestication as a whole, and how both human and environmental pressures have shaped the evolution of dingoes.

Abbott, I (2002) Origin and spread of the cat, Felis catus, on mainland Australia, with a discussion of the magnitude of its early impact on native fauna. Wildlife Research 29(1), 51-74.

Abbott, I (2011) The importation, release, establishment, spread, and early impact on prey animals of the red fox Vulpes vulpes in Victoria and adjoining parts of south-eastern Australia. Australian Zoologist 35(3), 463-533.

Abbott, IJ, Peacock D, and Short J (2014) The new guard: the arrival and impacts of cats and foxes. In 'Carnivores of Australia : past, present and future.' (Ed. A Glen and C Dickman) pp. 69-104 (CSIRO Publishing: Collingwood, Vic, Australia)

Ballard, JWO, and Wilson LAB (2019) The Australian dingo: untamed or feral? Frontiers in Zoology 16(1), 2.

Fairfax, RJ (2019) Dispersal of the introduced red fox (Vulpes vulpes) across Australia. Biological Invasions 21(4), 1259-1268.

Field, MA, Yadav S, Dudchenko O, Esvaran M, Rosen BD, Skvortsova K, Edwards RJ, Keilwagen J, Cochran BJ, Manandhar B, Bustamante S, Rasmussen JA, Melvin RG, Chernoff B, Omer A, Colaric Z, Chan EKF, Minoche AE, Smith TPL, Gilbert MTP, Bogdanovic O, Zammit RA, Thomas T, Aiden EL, and Ballard JWO (2022) The Australian dingo is an early offshoot of modern breed dogs. Science Advances 8(16), eabm5944.

6. PLOS authors have the option to publish the peer review history of their article (what does this mean?). If published, this will include your full peer review and any attached files.

Reviewer #1: **Yes: **Pat Shipman

Reviewer #2: No

---

## [Author Response · Author response to Decision Letter 0]

8 May 2023

Reviewer #1 

We thank the reviewer for their kind and constructive feedback. We have addressed their points as follows (pages and lines refer to original MS submission, not revised one):

- Page 3, Lines 51-53: We agree with this statement in principle. However, with respect to this specific instance: as dingoes are very thousands of kilometres removed from the biogeographic source of their wild ancestor (an East Asian wolf) it seems reasonable to assume that their ancestors were transported in a domesticated state. The only alternative explanation is a series of stepwise maritime translocations of wild dog stock throughout Mainland and Island Southeast Asia, but there is no archaeological or palaeontological evidence that populations of such animals existed in antiquity - aside from in New Guinea but for which the same issue of distance of distance from the ancestral home range exists. We have edited this section to better reflect the above contentions.

- Page 6, Line 132: Spelling error corrected

- Page 11, Line 293: Grammatical error corrected

- Page 14, Line 313: Grammatical/spelling errors corrected

Reviewer #2 

We thank the reviewer for their kind and constructive feedback. We have addressed their points as follows (pages and lines refer to original MS submission, not revised one):

- Line 14: The reviewer’s comment that Jackson et al. in both instances argue for Canis familiaris is correct. We have modified this paragraph to reflect the debatable taxonomy of dingoes.

- Line 51: We have added reference to these in text; ultimately this source (Abbott) was already who the cited papers’ arguments mostly rely on, but we have now cited his works directly in text along with Fairfax 2019. 

- Line 88: The idea of tamed vs untamed/domestic vs feral sensu Ballard and Wilson relies on a lack of definitive evidence for dingoes being domesticated in the past, hence “tamed” and “untamed” being safer designations. However, this contention is based on the interpretation of the biology/genetics/behaviour of modern wild dingoes which have experienced natural selection for millennia and thus may have lost many indicators of domestic heritage (whilst others remain such as gross cranial morphology). We have added some text to reflect this contention.

- Lines 525-542: We agree that the debate over the taxonomic status of dingoes involves both discussions of domesticity as well as morphological/genetic/behavioural differentiation, but ultimately those opinions on domesticity are derived from observations of morphology, genetic and behavioural characters. We have added another paragraph to convey this.

Academic Editor

Our manuscript has been edited and formatted to address the points raised by the reviewers. In addition, we have addressed your requests regarding formatting of the text and headings, figure and table captions, their in-text citations, and corresponding file names. We have also included within the acknowledgements section a statement concerning the obtainment of permits to carry out the research.

---

## [Editor Report · Decision Letter 1]

19 May 2023

Dingoes, companions in life and death: the significance of archaeological canid burial practices in Australia

PONE-D-23-07642R1

Dear Dr. Koungoulos,

We’re pleased to inform you that your manuscript has been judged scientifically suitable for publication and will be formally accepted for publication once it meets all outstanding technical requirements.

Kind regards,

Michael D. Petraglia, Ph.D.

Academic Editor

PLOS ONE
---

## [Editor Report · Acceptance letter]

24 May 2023

PONE-D-23-07642R1 

Dingoes, companions in life and death: the significance of archaeological canid burial practices in Australia 

Dear Dr. Koungoulos:

I'm pleased to inform you that your manuscript has been deemed suitable for publication in PLOS ONE. Congratulations! Your manuscript is now with our production department. 

Kind regards, 

on behalf of

Professor Michael D. Petraglia 

Academic Editor

PLOS ONE